# Quantum Algorithms For Deep Convolutional Neural Network

**Iordanis Kerenidis, Jonas Landman & Anupam Prakash**
Institut de Recherche en Informatique Fondamentale (IRIF)
Université de Paris, CNRS
Paris, France
`landman@irif.fr`

## Abstract

Quantum computing is a powerful computational paradigm with applications in several fields, including machine learning. In the last decade, deep learning, and in particular Convolutional Neural Networks (CNN), have become essential for applications in signal processing and image recognition. Quantum deep learning, however, remains a challenging problem, as it is difficult to implement non linearities with quantum unitaries Schuld et al. (2014). In this paper we propose a quantum algorithm for evaluating and training deep convolutional neural networks with potential speedups over classical CNNs for both the forward and backward passes. The quantum CNN (QCNN) reproduces completely the outputs of the classical CNN and allows for non linearities and pooling operations. The QCNN is in particular interesting for deep networks and could allow new frontiers in the image recognition domain, by allowing for many more convolution kernels, larger kernels, high dimensional inputs and high depth input channels. We also present numerical simulations for the classification of the MNIST dataset to provide practical evidence for the efficiency of the QCNN.

## 1 Introduction

The growing importance of deep learning in research, in industry and in our society will require extreme computational power as the dataset sizes and the complexity of these algorithms is expected to increase. Quantum computers are a good candidate to answer this challenge. The recent progress in the physical realization of quantum processors and the advances in quantum algorithms increases the importance of understanding their capabilities and limitations. In particular, the field of quantum machine learning has witnessed many innovative algorithms that offer speedups over their classical counterparts Kerenidis et al. (2019); Lloyd et al. (2013; 2014); Kerenidis & Prakash (2017b); Wiebe et al. (2014a).

Quantum deep learning refers to the problem of creating quantum circuits that mimic and enhance the operations of neural networks. It has been studied in several works Allcock et al. (2018); Rebentrost et al. (2018); Wiebe et al. (2014b) but remains challenging as it is difficult to implement non linearities with quantum unitaries Schuld et al. (2014). In this work we propose a quantum algorithm for convolutional neural networks (CNN), a type of deep learning designed for visual recognition, signal processing and time series. We also provide results of numerical simulations to evaluate the running time and accuracy of the quantum convolutional neural network (QCNN). Note that our algorithm is theoretical and could be compiled on any type of quantum computers (trapped ions, superconducting qubits, cold atoms, photons, etc.)

The CNN was originally developed by LeCun et al. (1998) in the 1980's. They have achieved great practical success over the last decade Krizhevsky et al. (2012) and have been used in cutting-edge domains like autonomous cars Bojarski et al. (2016) and gravitational wave detection George & Huerta (2018). Despite these successes, CNNs suffer from computational bottlenecks due to the size of the optimization space and the complexity of the inner operations, these bottlenecks make deep CNNs resource expensive.

The growing interest in quantum machine learning has led researchers to develop different variants of Quantum Neural Networks (QNN). The quest for designing quantum analogs of neural networks is challenging due to the modular layer architecture of the neural networks and the presence of non linearities, pooling, and other non unitary operations, as explained in Schuld et al. (2014). Several strategies have been tried in order to to implement some features of neural networks Allcock et al. (2018); Wiebe et al. (2014b); Beer et al. (2019) in the quantum setting.

Variational quantum circuits provide another path to the design of QNNs, this approach has been developed in Farhi & Neven (2018); Henderson et al. (2019); Killoran et al. (2018). A quantum convolutional neural network architecture using variational circuits was recently proposed Cong et al. (2018). However further work is required to provide evidence that such techniques can outperform classical neural networks in machine learning settings.

## 2 PRELIMINARIES

### 2.1 CONVOLUTION PRODUCT AS MATRIX MULTIPLICATION

We briefly introduce the formalism and notation concerning classical convolution product and its equivalence with matrix multiplication. More details can be found in Appendix (Section C). A single layer $\ell$ of the classical CNN does the following operations: from an input image $X^\ell \in \mathbb{R}^{H^\ell \times W^\ell \times D^\ell}$ seen as a 3D tensor, and a kernel $K^\ell \in \mathbb{R}^{H \times W \times D^\ell \times D^{\ell+1}}$ seen as a 4D tensor, it performs a convolution product and outputs $X^{\ell+1} = X^\ell * K^\ell$, with $X^{\ell+1} \in \mathbb{R}^{H^{\ell+1} \times W^{\ell+1} \times D^{\ell+1}}$. This convolution operation is equivalent to the matrix multiplication $A^\ell F^\ell = Y^{\ell+1}$ where $A^\ell, F^\ell$ and $Y^{\ell+1}$ are suitably vectorized versions of $X^\ell, K^\ell$ and $X^{\ell+1}$ respectively. The output of the layer $\ell$ of the CNN is $f(X^{\ell+1})$ where $f$ is a non linear function.

### 2.2 QUANTUM COMPUTING

For a detailed introduction to quantum computing and its applications to machine learning in the context of this work, we invite the reader to look at Appendix F. We also refer to Nielsen & Chuang (2002b) for a more complete overview of quantum computing.

In this part we will discuss only briefly the core notions of quantum computing. Like a classical bit, a quantum bit (qubit) can be $|0\rangle$, $|1\rangle$, but can also be in a superposition state $\alpha |0\rangle + \beta |1\rangle$ with amplitudes $(\alpha, \beta) \in \mathbb{C}$ such that $|\alpha|^2 + |\beta|^2 = 1$. With $n$ qubits it is then possible to construct a superposition of the $2^n$ binary combinations possible, each with a specific amplitude. We will note the $i^{th}$ combination (e.g. $|01\cdots110\rangle$) as $|i\rangle$. A vector $v \in \mathbb{R}^d$ can be encoded in a quantum state made of $\lceil \log(d) \rceil$ qubits. This encoding is a quantum superposition, where the components $(v_1, \cdots, v_d)$ of $v$ are used as the amplitudes of the $d$ binary combinations. We note this state $|v\rangle := \frac{1}{\|v\|} \sum_{i \in [d]} v_i |i\rangle$, where $|i\rangle$ is a register representing the $i^{th}$ vector in the standard basis.

Quantum computation proceeds by applying quantum gates which are defined to be unitary matrices acting on 1 or 2 qubits, for example the Hadamard gate that maps $|0\rangle \mapsto \frac{1}{\sqrt{2}}(|0\rangle + |1\rangle)$ and $|1\rangle \mapsto \frac{1}{\sqrt{2}}(|0\rangle - |1\rangle)$. The output of the computation is a quantum state that can be measured to obtain classical information. The measurement of a qubit $\alpha |0\rangle + \beta |1\rangle$ yields either 0 or 1, with probability equal to the square of the respective amplitude. A detailed discussion of the results from quantum machine learning and linear algebra used in this work can be found in Appendix (Section F).

## 3 MAIN RESULTS

In this paper, we design a quantum convolutional neural network (QCNN) algorithm with a modular architecture that allows for any number of layers, any number and size of kernels, and that can support a large variety of non linearity and pooling methods. Our main technical contributions include a new notion of a quantum convolution product, the development of a quantum sampling technique well suited for information recovery in the context of CNNs and a proposal for a quantum backpropagation algorithm for efficient training of the QCNN.

The QCNN can be directly compared to the classical CNN as it has the same inputs and outputs. We show that it offers a speedup compared to certain cases of classical CNN for both the forward pass and for training using backpropagation. For each layer, on the forward pass (Algorithm 1), the speedup is exponential in the size of the layer (number of kernels) and almost quadratic on the spatial dimension of the input. We next state informally the speedup for the forward pass, the formal version appears as Theorem D.1.

**Result 1** *(Quantum Convolution Layer)*
*Let $X^\ell$ be the input and $K^\ell$ be the kernel for layer $\ell$ of a convolutional neural network, and $f$ : $\mathbb{R} \mapsto [0, C]$ with $C > 0$ be a non linear function so that $f(X^{\ell+1}) := f(X^\ell * K^\ell)$ is the output for layer $\ell$. Given $X^\ell$ and $K^\ell$ stored in QRAM (Quantum Random Access Memory), there is a quantum algorithm that, for precision parameters $\epsilon > 0$ and $\eta > 0$, creates quantum state $|f(\overline{X}^{\ell+1})\rangle$ such that $\left\| f(\overline{X}^{\ell+1}) - f(X^{\ell+1}) \right\|_\infty \leq 2\epsilon$ and retrieves classical tensor $\mathcal{X}^{\ell+1}$ such that for each pixel $j$,*

$$\begin{cases} |\mathcal{X}_j^{\ell+1} - f(X_j^{\ell+1})| \leq 2\epsilon & if \quad f(\overline{X}_j^{\ell+1}) \geq \eta \\ \mathcal{X}_j^{\ell+1} = 0 & if \quad f(\overline{X}_j^{\ell+1}) < \eta \end{cases} \tag{1}$$

*The running time of the algorithm is $\widetilde{O}\left( \frac{1}{\epsilon\eta^2} \cdot \frac{M\sqrt{C}}{\sqrt{\mathbb{E}(f(\overline{X}^{\ell+1}))}} \right)$ where $\mathbb{E}(\cdot)$ represents the average value, $\widetilde{O}$ hides factors poly-logarithmic in the size of $X^\ell$ and $K^\ell$ and the parameter $M = \max_{p,q} \|A_p\| \|F_q\|$ is the maximum product of norms from subregions of $X^\ell$ and $K^\ell$.*

We see that the number of elements in the input and the kernels appear only with a poly-logarithmic contribution in the running time. This is one of the main advantages of our algorithm and it allows us to use for larger and even exponentially deeper kernels. For the number of elements in the input, their number is hidden in the precision parameter $\eta$ in the running time. Indeed, a sufficiently large fraction of pixels must be sampled from the output of the quantum convolution to retrieve the meaningful information. In the Numerical Simulations (Section 6) we provide empirical estimates for $\eta$. For details about the QRAM, see Appendix F.2.

Following the forward pass, a loss function $\mathcal{L}$ is computed for the output of a classical CNN. The backpropagation algorithm is then used to calculate, layer by layer, the gradient of this loss with respect to the elements of the kernels $K^\ell$, in order to update them through gradient descent. We state our quantum backpropagation algorithm next, the formal version of this result appears as Theorem E.1

**Result 2** *(Quantum Backpropagation for Quantum CNN)*
*Given the forward pass quantum algorithm in Result 1, and given the kernel matrix $F^\ell$, input matrices $A^\ell$ and $Y^\ell$, stored in the QRAM for each layer $\ell$, and a loss function $\mathcal{L}$, there is a quantum backpropagation algorithm that estimates each element of the gradient tensor $\frac{\partial \mathcal{L}}{\partial F^\ell}$ within additive error $\delta \left\| \frac{\partial \mathcal{L}}{\partial F^\ell} \right\|$, and updates $F^\ell$ according to a gradient descent update rule. The running time of a single layer $\ell$ for quantum backpropagation is given by*

$$O\left( \left( \left( \mu(A^\ell) + \mu(\frac{\partial \mathcal{L}}{\partial Y^{\ell+1}}) \right) \kappa(\frac{\partial \mathcal{L}}{\partial F^\ell}) + \left( \mu(\frac{\partial \mathcal{L}}{\partial Y^{\ell+1}}) + \mu(F^\ell) \right) \kappa(\frac{\partial \mathcal{L}}{\partial Y^\ell}) \right) \frac{\log 1/\delta}{\delta^2} \right) \tag{2}$$

*where for a matrix $V \in \mathbb{R}^{n \times n}$, $\kappa(V)$ is the condition number and $\mu(V) \leq \sqrt{n}$ is a matrix dependent parameter defined in Equation (5).*

For the quantum back-propagation algorithm, we introduce a quantum tomography algorithm with $\ell_\infty$ norm guarantees, that could be of independent interest. It is exponentially faster than tomography with $\ell_2$ norm guarantees and is given as Theorem G.1 in Section G. Numerical simulations on classifying the MNIST dataset show that our quantum CNN achieves a similar classification accuracy as the classical CNN.

## 4   FORWARD PASS FOR QCNN

The forward pass algorithm for the QCNN implements the quantum analog of a single quantum convolutional layer. It includes a convolution product between an input and a kernel, followed by

the application of a non linear function and pooling operations to prepare the next layer's input. We provide an overview of the main ideas of the algorithm here, the complete technical details are given in the Appendix (Section D).

---

**Algorithm 1** QCNN Layer

---

**Require:** Matrix $A^\ell$ representing input to layer $\ell$ and kernel matrix $F^\ell$ stored in QRAM. Precision parameters $\epsilon$ and $\eta$, a boolean circuit for a non linear function $f : \mathbb{R} \mapsto [0, C]$.

**Ensure:** Outputs the data matrix $A^{\ell+1}$ for the next layer which is the result of the convolution between the input and the kernel, followed by a non linearity and pooling.

---

1: **Step 1: Quantum Convolution**
   **1.1: Inner product estimation**
   Perform the following mapping, using QRAM queries on rows $A^\ell_p$ and columns $F^\ell_q$, followed by quantum inner product estimation (Theorems F.2 and F.4) to implement the mapping
   $\frac{1}{K} \sum_{p,q} |p\rangle |q\rangle \mapsto \frac{1}{K} \sum_{p,q} |p\rangle |q\rangle |\overline{P}_{pq}\rangle |g_{pq}\rangle$
   Where $\overline{P}_{pq}$ is $\epsilon$-close to $P_{pq} = \frac{1+\langle A^\ell_p | F^\ell_q \rangle}{2}$, $K = \sqrt{H^{\ell+1} W^{\ell+1} D^{\ell+1}}$ is a normalisation factor and $|g_{pq}\rangle$ is some garbage quantum state that can be ignored.
   **1.2: Non linearity**
   Use an arithmetic circuit and two QRAM queries to obtain $\overline{Y}^{\ell+1}$, an $\epsilon$-approximation of the convolution output $Y^{\ell+1}_{p,q} = (A^\ell_p, F^\ell_q)$ and apply the non linear function $f$ as a boolean circuit to obtain $\frac{1}{K} \sum_{p,q} |p\rangle |q\rangle |f(\overline{Y}^{\ell+1}_{p,q})\rangle |g_{pq}\rangle$.

2: **Step 2: Quantum Sampling**
   Use Conditional Rotation and Amplitude Amplification to encode the values $\alpha'_{pq} := \frac{f(\overline{Y}^{\ell+1}_{pq})}{C}$ into the amplitudes to obtain $\frac{1}{K} \sum_{p,q} \alpha'_{pq} |p\rangle |q\rangle |f(\overline{Y}^{\ell+1}_{pq})\rangle |g_{pq}\rangle$. Perform $\ell_\infty$ tomography from Theorem G.1 with precision $\eta$, and obtain classically all positions and values $(p, q, f(\overline{Y}^{\ell+1}_{pq}))$ such that, with high probability, values above $\eta$ are known exactly, while others are set to 0.

3: **Step 3: QRAM Update and Pooling**
   Update the QRAM for the next layer $A^{\ell+1}$ while sampling. The implementation of pooling (Max, Average, etc.) can be done by a specific update to the QRAM data structure described in Section D.2.2.

---

In this algorithm, we propose the first quantum algorithm for performing the convolution product. Our algorithm is based on the observation that the convolution product can be regarded as a matrix product between reshaped matrices. The reshaped input's rows $A^\ell_p$ and the reshaped kernel's columns $F^\ell_q$ are loaded as quantum states, in superposition. Then the entries of the convolution $\langle A^\ell_p | F^\ell_q \rangle$ are estimated using a simple quantum circuit for inner product estimation and stored in an auxiliary register as in Step 1.1 of Algorithm 1.

One of the difficulties in the design of quantum neural networks is that non linear functions are hard to implement as unitary operations. We get around this difficulty by applying the non-linear function $f$ as a boolean circuit to the output of the quantum inner product estimation circuit in Step 1.2 of Algorithm 1. Most of the non linear functions in the machine learning literature can be implemented using small sized boolean circuits, our algorithm thus allows for many possible choices of the non-linear function $f$ (see Appendix F.1 for details on non linear boolean circuits in quantum circuits).

Step 2 of Algorithm 1 develops a quantum importance sampling procedure wherein the pixels with high values of $f(\overline{Y}^{\ell+1}_{pq})$ are read out with higher probability. This is done by encoding these values into the amplitudes of the quantum state using the well known Amplitude Amplification algorithm Brassard et al. (2002). This kind of importance sampling is a task that can be performed easily in the quantum setting and has no direct classical analog. Although it does not lead to asymptotic improvements for the algorithms running time, it could lead to improvements that are significant in practice.

More precisely, during the measurement of a quantum register in superposition, only one of its values appears, with a probability corresponding the the square of its amplitude. It implies that the

output's pixels measured with more probability are the ones with the highest value $f(Y_{p,q}^{\ell+1})$. Once measured, we read directly from the registers the position $p, q$ and the value itself. Thus we claim that we measure only a fraction of the quantum convolution product output, and that the set of pixels measured collect most of the meaningful information for the CNN, the other pixels being set to 0.

After being measured, each pixel's value and position are stored in a QRAM to be used as quantum state for next layer's input. During this phase, it is possible to discard or aggregate some values to perform pooling operations as described in Step 3 of Algorithm 1. The forward pass for the QCNN thus includes the the convolution product, the non linearity $f$ and pooling operation, in time poly-logarithmic in the kernel's dimensions. In comparison, the classical CNN layer in linear in both kernel and input dimensions.

Note finally that quantum importance sampling in Step 2 implies that the non linear function $f$ be bounded by a parameter $C > 1$. In our experiments we use the capReLu function, which is a modified ReLu function that becomes constant above $C$.

## 5 QUANTUM BACKPROPAGATION ALGORITHM

The second algorithm required for the QCNN is the quantum backpropagation algorithm given as Algorithm 2. Like the classical backpropagation algorithm, it updates all kernels weights according to the derivatives of a given loss function $\mathcal{L}$. In this sectiion, we explain the main ideas and compare it to the classical backpropagation algorithm, the complete details are given in Appendix (Section E).

---

**Algorithm 2** Quantum Backpropagation

---

**Require:** Precision parameter $\delta$. Data matrices $A^\ell$ and kernel matrices $F^\ell$ stored in QRAM for each layer $\ell$.
**Ensure:** Outputs gradient matrices $\frac{\partial \mathcal{L}}{\partial F^\ell}$ and $\frac{\partial \mathcal{L}}{\partial Y^\ell}$ for each layer $\ell$.

1: Calculate the gradient for the last layer $L$ using the outputs and the true labels: $\frac{\partial \mathcal{L}}{\partial Y^L}$
2: **for** $\ell = L - 1, \cdots, 0$ **do**
3:     **Step 1 : Modify the gradient**
    With $\frac{\partial \mathcal{L}}{\partial Y^{\ell+1}}$ stored in QRAM, set to 0 some of its values to take into account pooling, tomography and non linearity that occurred in the forward pass of layer $\ell$. These values correspond to positions that haven't been sampled nor pooled, since they have no impact on the final loss.

4:     **Step 2 : Matrix-matrix multiplications**
    With the modified values of $\frac{\partial \mathcal{L}}{\partial Y^{\ell+1}}$, use quantum linear algebra algorithm (Theorem F.7) to perform the matrix-matrix multiplications $(A^\ell)^T \cdot \frac{\partial L}{\partial Y^{\ell+1}}$ and $\frac{\partial \mathcal{L}}{\partial Y^{\ell+1}} \cdot (F^\ell)^T$, allowing to obtain quantum states corresponding to $\frac{\partial \mathcal{L}}{\partial F^\ell}$ and $\frac{\partial \mathcal{L}}{\partial Y^\ell}$.
5:     **Step 3 : $\ell_\infty$ tomography**
    Measure the previous outputs, as in Algorithm 3. This allows to estimate each entry of $\frac{\partial \mathcal{L}}{\partial F^\ell}$ and $\frac{\partial \mathcal{L}}{\partial Y^\ell}$ with errors $\delta \left\| \frac{\partial \mathcal{L}}{\partial F^\ell} \right\|$ and $\delta \left\| \frac{\partial \mathcal{L}}{\partial Y^\ell} \right\|$ respectively, using $\ell_\infty$ tomography from Theorem G.1. Store all elements of $\frac{\partial \mathcal{L}}{\partial F^\ell}$ in QRAM.
6:     **Step 4 : Gradient descent**
    From the previous tomography, perform the gradient descent to update the values of $F^\ell$ in QRAM: $F_{s,q}^\ell \leftarrow F_{s,q}^\ell - \lambda \left( \frac{\partial \mathcal{L}}{\partial F_{s,q}^\ell} \pm 2\delta \left\| \frac{\partial \mathcal{L}}{\partial F^\ell} \right\|_2 \right)$.
7: **end for**

---

We describe briefly detail the implementation of quantum backpropagation at layer $\ell$. The algorithm assumes that $\frac{\partial \mathcal{L}}{\partial Y^{\ell+1}}$ is known. First, the backpropagation of the quantum convolution product is equivalent to the classical one, and we use the matrix-matrix multiplication formulation to obtain the derivatives $\frac{\partial \mathcal{L}}{\partial F^\ell}$ and $\frac{\partial \mathcal{L}}{\partial Y^\ell}$. The first one is the result wanted and the second one is needed for layer $\ell - 1$. This matrix-matrix multiplication can be implemented as a quantum circuit, by decomposing into several matrix-vector multiplications, known to be efficient, with a running time depending on the ranks and Frobenius norm of the matrices. We obtain a quantum state corresponding to a

superposition of all derivatives. We use again the $\ell_\infty$ tomography to retrieve each derivative with precision $\delta > 0$ such that, for all kernel's weight $F_{s,q}^\ell$ we have approximated it's loss derivative with $\overline{\frac{\partial \mathcal{L}}{\partial F_{s,q}^\ell}}$, with an error bounded by $\left| \frac{\partial \mathcal{L}}{\partial F_{s,q}^\ell} - \overline{\frac{\partial \mathcal{L}}{\partial F_{s,q}^\ell}} \right| \leq 2\delta \left\| \frac{\partial \mathcal{L}}{\partial F^\ell} \right\|_2$. This implies that the gradient descent rule is perturbed by $2\delta \left\| \frac{\partial \mathcal{L}}{\partial F^\ell} \right\|_2$ at most, see Appendix (Section E.4).

We also take into account the effects of quantum non linearity, quantum measurement and pooling. The quantum pooling operation is equivalent to the classical one, where pixels that were not selected during pooling see their derivative set to 0. Quantum measurement is similar, since pixels that haven't been measured don't contribute to the gradient. For the non linearity, as in the classical case, pixels with negative values were set to zero, hence should have no contribution to the gradient. Additionally, because we used the capReLu function, pixels bigger than the threshold $C$ must also have null derivatives. This two rules can be implemented by combining them with measurement rules compared to classical backpropagation, see Appendix (Section E.2.2) for details.

## 6 NUMERICAL SIMULATIONS

As described above, the adaptation of the CNNs to the quantum setting implies some modifications that could alter the efficiency of the learning or classifying phases. We now present some experiments to show that such modified CNNs can converge correctly, as the original ones.

The experiment, using the PyTorch library developed by Paszke et al. (2017), consists of training classically a small convolutional neural network for which we have added a "quantum" sampling after each convolution. Instead of parametrising it with the precision $\eta$, we have chosen to use the sampling ratio $\sigma$ that represents the fraction of pixels drawn during tomography. This two definitions are equivalent, as shown in Appendix (Section D.1.5), but the second one is more intuitive regarding the running time and the simulations.

We also add a noise simulating the amplitude estimation (parameter $\epsilon$), followed by a capReLu instead of the usual ReLu (parameter $C$), and a noise during the backpropagation (parameter $\delta$). In the following results, we observe that our quantum CNN is able to learn and classify visual data from the widely used MNIST dataset. This dataset is made of 60.000 training images and 10.000 testing images of handwritten digits. Each image is a 28x28 grayscale pixels between 0 and 255 (8 bits encoding), before normalization.

Let's first observe the "quantum" effects on an image of the dataset. In particular, the effect of the capped non linearity, the introduction of noise and the quantum sampling.

We now present the full simulation of our quantum CNN. In the following, we use a simple network made of 2 convolution layers, and compare our quantum CNN to the classical one. The first and second layers are respectively made of 5 and 10 kernels, both of size 7x7. A three-layer fully connected network is applied at the end and a softmax activation function is applied on the last layer to detect the predicted outcome over 10 classes (the ten possible digits). Note that we didn't introduce pooling, being equivalent between quantum and classical algorithms and not improving the results on our CNN. The objective of the learning phase is to minimize the loss function, defined by the negative log likelihood of the classification on the training set. The optimizer used was a built-in Stochastic Gradient Descent.

Using PyTorch, we have been able to implement the following quantum effects (the first three points are shown in Figure 1):

- The addition of a noise, to simulate the approximation of amplitude estimation during the forward quantum convolution layer, by adding gaussian noise centered on 0 and with standard deviation $2M\epsilon$, with $M = \max_{p,q} \|A_p\| \|F_q\|$.

- A modification of the non linearity: a ReLu function which is constant above the value $T$ (the cap).

- A sampling procedure to apply on a tensor with a probability distribution proportional to the tensor itself, reproducing the quantum sampling with ratio $\sigma$.

- The addition of a noise during the gradient descent, to simulate the quantum backpropagation, by adding a gaussian noise centered on 0 with standard deviation $\delta$, multiplied by the norm of the gradient, as given by Equation (28).

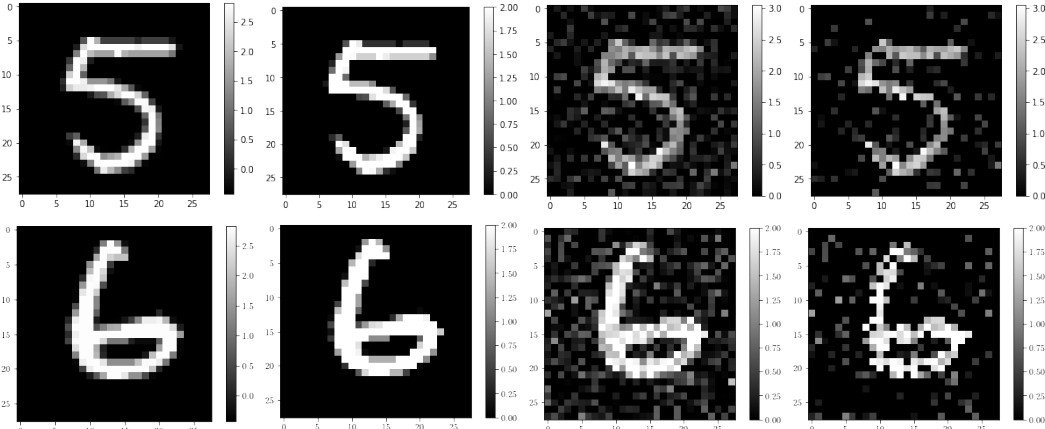

Figure 1: Effects of the QCNN on a 28x28 input image. From left to right: original image, image after applying a capReLu activation function with a cap $C$ at 2.0, introduction of a strong noise during amplitude estimation with $\epsilon = 0.5$, quantum sampling with ratio $\sigma = 0.4$ that samples the highest values in priority. The useful information tends to be conserved in this example. The side gray scale indicates the value of each pixel. Note that during the QCNN layer, a convolution is supposed to happen before the last image but we chose not to perform it for visualisation matter.

The CNN used for this simulation may seem "small" compared to the standards AlexNet developed by Krizhevsky et al. (2012) or VGG-16 by Simonyan & Zisserman (2014), or those used in industry. However simulating this small QCNN on a classical computer was already very computationally intensive and time consuming, due to the "quantum" sampling task, apparently not optimized for a classical implementation in PyTorch. Every single training curve showed in Figure 9 could last for 4 to 8 hours. Hence adding more convolutional layers wasn't convenient. Similarly, we didn't compute the loss on the whole testing set (10.000 images) during the training to plot the testing curve. However we have computed the test losses and accuracies once the model trained (see Table 4), in order to detect potential overfitting cases.

We now present the result of the training phase for a quantum version of this CNN, where partial quantum sampling is applied, for different sampling ratio (number of samples taken from the resulting convolution). Since the quantum sampling gives more probability to observe high value pixels, we expect to be able to learn correctly even with small ratio ($\sigma \leq 0.5$). We compare these training curve to the classical one. The learning has been done on two epochs, meaning that the whole dataset is used twice. The following plots show the evolution of the loss $\mathcal{L}$ during the iterations on batches. This is the standard indicator of the good convergence of a neural network learning phase. We can compare the evolution of the loss between a classical CNN and our QCNN for different parameters. Most results are presented in Appendix (Section H).

Our simulations show that the QCNN is able to learn despite the introduction of noise, tensor sampling and other modifications. In particular it shows that only a fraction of the information is meaningful for the neural network, and that the quantum algorithm captures this information in priority. This learning can be more or less efficient depending on the choice of the key parameters. For decent values of these parameters, the QCNN is able to converge during the training phase. It can then classify correctly on both training and testing set, indicating neither overfitting nor underfitting.

We notice that the learning curves sometimes present a late start before the convergence initializes, in particular for small sampling ratio. This late start can be due to the random initialization of the kernel weights, that performs a meaningless convolution, a case where the quantum sampling of the output is of no interest. However it is very interesting to see that despite this late start, the kernel start converging once they have found a good combination.

Overall, it is possible that the QCNN presents some behaviors that have no classical equivalence. Understanding their potential effects, positive or negative, is an open question, all the more so as the effects of the classical CNN's hyperparameters are already a topic of active research, see the work of Samek et al. (2017) for details. Note also that the neural network used in this simulation is

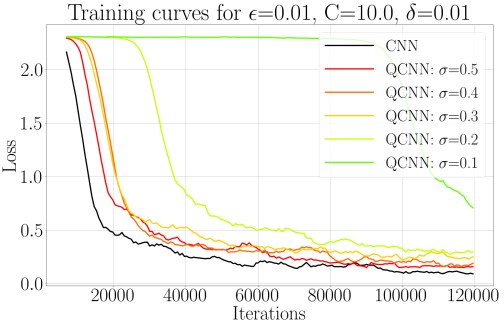

Figure 2: Training curves comparison between the classical CNN and the Quantum CNN (QCNN) for $\epsilon = 0.01$, $C = 10$, $\delta = 0.01$ and the sampling ratio $\sigma$ from 0.1 to 0.5. We can observe a learning phase similar to the classical one, even for a weak sampling of 20% or 30% of each convolution output, which tends to show that the meaningful information is distributed only at certain location of the images, coherently with the purpose of the convolution layer. Even for a very low sampling ratio of 10%, we observe a convergence despite a late start.

rather small. A following experiment would be to simulate a quantum version of a standard deeper CNN (AlexNet or VGG-16), eventually on more complex dataset, such as CIFAR-10 developed by Krizhevsky & Hinton (2009) or Fashion MNIST by Xiao et al. (2017).

## 7 CONCLUSIONS

We have presented a quantum algorithm for evaluating and training convolutional neural networks (CNN). At the core of this algorithm, we have developed a novel quantum algorithm for computing a convolution product between two tensors, with a substantial speed up. This technique could be reused in other signal processing tasks that could benefit from an enhancement by a quantum computer. Layer by layer, convolutional neural networks process and extract meaningful information. Following this idea of learning foremost important features, we have proposed a new approach of quantum tomography where the most meaningful information is sampled with higher probability, hence reducing the complexity of our algorithm.

Our QCNN is complete in the sense that almost all classical architectures can be implemented in a quantum fashion: any (non negative and upper bounded) non linearity, pooling, number of layers and size of kernels are available. Our circuit is shallow and could be run on relatively small quantum computers. One could repeat the main loop many times on the same shallow circuit, since performing the convolution product is simple, and is similar for all layer. The pooling and non linearity are included in the loop. Our building block approach, layer by layer, allows high modularity, and can be combined with work on quantum feedforward neural network developed by Allcock et al. (2018).

The running time presents a speedup compared to the classical algorithm, due to fast linear algebra when computing the convolution product, and by only sampling the important values from the resulting quantum state. This speedup can be highly significant in cases where the number of channels $D^{\ell}$ in the input tensor is high (high dimensional time series, videos sequences, games play) or when the number of kernels $D^{\ell+1}$ is big, allowing deep architectures for CNN, which was the case in the recent breakthrough of DeepMind AlphaGo algorithm of Silver et al. (2016). The Quantum CNN also allows larger kernels, that could be used for larger input images, since the size the kernels must be a contant fraction of the input in order to recognize patterns. However, despite our new techniques to reduce the complexity, applying a non linearity and reusing the result of a layer for the next layer make register encoding and state tomography mandatory, hence preventing from having an exponential speedup on the number of input parameters.

Finally we have presented a backpropagation algorithm that can also be implemented as a quantum circuit. The numerical simulations on a small CNN show that despite the introduction of noise and sampling, the QCNN can efficiently learn to classify visual data from the MNIST dataset, performing a similar accuracy than the classical CNN.

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

## APPENDIX A    VARIABLE SUMMARY

We recall the most important variables for layer $\ell$. They represent tensors, their approximations, and their reshaped versions.

| Data | Variable | Dimensions | Indices |
|------|----------|------------|---------|
| Input | $X^\ell$ | $H^\ell \times W^\ell \times D^\ell$ | $(i^\ell, j^\ell, d^\ell)$ |
|  | $Y^\ell$ | $(H^\ell W^\ell) \times D^\ell$ | - |
|  | $A^\ell$ | $(H^{\ell+1} W^{\ell+1}) \times (HWD^\ell)$ | $(p, r)$ |
| Kernel | $K^\ell$ | $H \times W \times D^\ell \times D^{\ell+1}$ | $(i, j, d, d')$ |
|  | $F^\ell$ | $(HWD^\ell) \times D^{\ell+1}$ | $(s, q)$ |

Table 1: Summary of input variables for the $\ell^{th}$ layer, along with their meaning, dimensions and corresponding notations. These variables are common for both *quantum* and *classical* algorithms. We have omitted indices for $Y^\ell$ which don't appear in our work.

| Data | Variable | Dimensions | Indices |
|------|----------|------------|---------|
| Output of Quantum Convolution | $f(\overline{Y}^{\ell+1})$ | $(H^{\ell+1} W^{\ell+1}) \times D^{\ell+1}$ | $(p, q)$ |
|  | $f(\overline{X}^{\ell+1})$ | $H^{\ell+1} \times W^{\ell+1} \times D^{\ell+1}$ | $(i^{\ell+1}, j^{\ell+1}, d^{\ell+1})$ |
| Output of Quantum Tomography | $\mathcal{X}^{\ell+1}$ | $H^{\ell+1} \times W^{\ell+1} \times D^{\ell+1}$ | $(i^{\ell+1}, j^{\ell+1}, d^{\ell+1})$ |
| Output of Quantum Pooling | $\tilde{\mathcal{X}}^{\ell+1}$ | $\frac{H^{\ell+1}}{P} \times \frac{W^{\ell+1}}{P} \times D^{\ell+1}$ | $(\tilde{i}^{\ell+1}, \tilde{j}^{\ell+1}, \tilde{d}^{\ell+1})$ |

Table 2: Summary of variables describing outputs of the layer $\ell$, with the *quantum* algorithm.

| Data | Variable | Dimensions | Indices |
|------|----------|------------|---------|
| Output of Classical Convolution | $f(Y^{\ell+1})$ | $(H^{\ell+1} W^{\ell+1}) \times D^{\ell+1}$ | $(p, q)$ |
|  | $f(X^{\ell+1})$ | $H^{\ell+1} \times W^{\ell+1} \times D^{\ell+1}$ | $(i^{\ell+1}, j^{\ell+1}, d^{\ell+1})$ |
| Output of Classical Pooling | $\tilde{X}^{\ell+1}$ | $\frac{H^{\ell+1}}{P} \times \frac{W^{\ell+1}}{P} \times D^{\ell+1}$ | $(\tilde{i}^{\ell+1}, \tilde{j}^{\ell+1}, \tilde{d}^{\ell+1})$ |

Table 3: Summary of variables describing outputs of the layer $\ell$, with the *classical* algorithm.

Classical and quantum algorithms can be compared with these two diagrams:

$$
\begin{cases}
\text{Quantum convolution layer}: X^\ell \to |\overline{X}^{\ell+1}\rangle \to |f(\overline{X}^{\ell+1})\rangle \to \mathcal{X}^{\ell+1} \to \tilde{\mathcal{X}}^{\ell+1} \\
\text{Classical convolution layer}: X^\ell \to X^{\ell+1} \to f(X^{\ell+1}) \to \tilde{X}^{\ell+1}
\end{cases}
\tag{3}
$$

We finally provide some remarks that could clarify some notations ambiguity:

- Formally, the output of the quantum algorithm is $\tilde{\mathcal{X}}^{\ell+1}$. It is used as input for the next layer $\ell + 1$. But we consider that all variables' names are *reset* when starting a new layer: $X^{\ell+1} \leftarrow \tilde{\mathcal{X}}^{\ell+1}$.

- For simplicity, we have sometimes replaced the indices $(i^{\ell+1}, j^{\ell+1}, d^{\ell+1})$ by $n$ to index the elements of the output.

- In Section D.2.2, the input for layer $\ell + 1$ is stored as $A^{\ell+1}$, for which the elements are indexed by $(p', r')$.

## APPENDIX B    PRELIMINARIES IN QUANTUM INFORMATION

We introduce a basic and broad-audience quantum information background necessary for this work. For a more detailed introduction we recommend Nielsen & Chuang (2002a).

## B.1  QUANTUM INFORMATION

**Quantum Bits and Quantum Registers:**  The bit is the most basic unit of classical information. It can be either in state 0 or 1. Similarly a quantum bit or *qubit*, is a quantum system that can be is state $|0\rangle$, $|1\rangle$ (the *braket* notation $|\cdot\rangle$ is a reminder that the bit considered is a quantum system) or in superposition of both states $\alpha |0\rangle + \beta |1\rangle$ with coefficients $\alpha, \beta \in \mathbb{C}$ such that $|\alpha|^2 + |\beta|^2 = 1$. The *amplitudes* $\alpha$ and $\beta$ are linked to the probabilities of observing either 0 or 1 when *measuring* the qubit, since $P(0) = |\alpha|^2$ and $P(1) = |\beta|^2$.

Before the measurement, any superposition is possible, which gives quantum information special abilities in terms of computation. With $n$ qubits, the $2^n$ possible binary combinations can exist simultaneously, each with a specific amplitude. For instance we can consider an uniform distribution $\frac{1}{\sqrt{n}} \sum_{i=0}^{2^n-1} |i\rangle$ where $|i\rangle$ represents the $i^{th}$ binary combination (e.g. $|01 \cdots 1001\rangle$). Multiple qubits together are often called a *quantum register*.

In its most general formulation, a quantum state with $n$ qubits can be seen as vector in a complex Hilbert space of dimension $2^n$. This vector must be normalized under $\ell_2$-norm, to guarantee that the squared amplitudes sum to 1.

**Quantum Computation:**  To process qubits and therefore quantum registers, we use quantum gates. These gates are *unitary operators* in the Hilbert space as they should map unit-norm vectors to unit-norm vectors. Formally, we can see a quantum gate acting on $n$ qubits as a matrix $U \in \mathbb{C}^{2^n}$ such that $UU^\dagger = U^\dagger U = I$, where $U^\dagger$ is the conjugate transpose of $U$. Some basic single qubit gates includes the NOT gate $\begin{pmatrix} 0 & 1 \\ 1 & 0 \end{pmatrix}$ that inverts $|0\rangle$ and $|1\rangle$, or the Hadamard gate $\frac{1}{\sqrt{2}} \begin{pmatrix} 1 & 1 \\ 1 & -1 \end{pmatrix}$ that maps $|0\rangle \mapsto \frac{1}{\sqrt{2}}(|0\rangle + |1\rangle)$ and $|1\rangle \mapsto \frac{1}{\sqrt{2}}(|0\rangle - |1\rangle)$, creating the quantum superposition.

Finally, multiple qubits gates exist, such as the Controlled-NOT that applies a NOT gate on a target qubit conditioned on the state of a control qubit.

The main advantage of quantum gates is their ability to be applied to a superposition of inputs. Indeed, given a gate $U$ such that $U |x\rangle \mapsto |f(x)\rangle$, we can apply it to all possible combinations of $x$ at once $U(\frac{1}{C} \sum_x |x\rangle) \mapsto \frac{1}{C} \sum_x |f(x)\rangle$.

We now state some primitive quantum circuits, which we will use in our algorithm: For two integers $i$ and $j$, we can check their equality with the mapping $|i\rangle |j\rangle |0\rangle \mapsto |i\rangle |j\rangle |[i = j]\rangle$. For two real value numbers $a > 0$ and $\delta > 0$, we can compare them using $|a\rangle |\delta\rangle |0\rangle \mapsto |a\rangle |\delta\rangle |[a \leq \delta]\rangle$. Finally, for a real value numbers $a > 0$, we can obtain its square $|a\rangle |0\rangle \mapsto |a\rangle |a^2\rangle$. Note that these circuits are basically a reversible version of the classical ones and are linear in the number of qubits used to encode the input values.

Any classical boolean function can be implemented in a quantum unitary, even though this seems at first contradictory with the requirements of unitaries (reversibility, linearity). Let $\sigma : \mathbb{R} \mapsto \mathbb{R}$ be a classical function, we define $U_\sigma$ the unitary that acts as $U_\sigma |x\rangle |0\rangle \mapsto |x\rangle |\sigma(x)\rangle$. Using a second quantum register to encode the result of the function, the properties of quantum unitaries are respected.

## B.2  QUANTUM SUBROUTINES FOR DATA ENCODING

Knowing some basic principles of quantum information, the next step is to understand how data can be efficiently encoded using quantum states. While several approaches could exist, we present the most common one called *amplitude encoding*, which leads to interesting and efficient applications.

Let $x \in \mathbb{R}^d$ be a vector with components $(x_1, \cdots, x_d)$. Using only $\lceil \log(d) \rceil$ qubits, we can form $|x\rangle$, the quantum state encoding $x$, given by $|x\rangle = \frac{1}{\|x\|} \sum_{j=0}^{d-1} x_j |j\rangle$. We see that the $j^{th}$ component $x_j$ becomes the amplitude of $|j\rangle$, the $j^{th}$ binary combination (or equivalently the $j^{th}$ vector in the standard basis). Each amplitude must be divided by $\|x\|$ to preserve the unit $\ell_2$-norm of $|x\rangle$.

Similarly, for a matrix $A \in \mathbb{R}^{n \times d}$ or equivalently for $n$ vectors $A_i$ for $i \in [n]$, we can express each row of $A$ as $|A_i\rangle = \frac{1}{\|A_i\|} \sum_{i=0}^{d-1} A_{ij} |j\rangle$.

We can now explain an important definition, the ability to have *quantum access* to a matrix. This will be a requirements for many algorithms.

**Definition 1** *[Quantum Access to Data]*
*We say that we have quantum access to a matrix $A \in \mathbb{R}^{n \times d}$ if there exist a procedure to perform the following mapping, for $i \in [n]$, in time $T$:*

- $|i\rangle |0\rangle \mapsto |i\rangle |A_i\rangle$

- $|0\rangle \mapsto \frac{1}{\|A\|_F} \sum_i \|A_i\| |i\rangle$

By using appropriate data structures the first mapping can be reduced to the ability to perform a mapping of the form $|i\rangle |j\rangle |0\rangle \mapsto |i\rangle |j\rangle |A_{ij}\rangle$. The second requirement can be replaced by the ability of performing $|i\rangle |0\rangle \mapsto |i\rangle |\|A_i\|\rangle$ or to just have the knowledge of each norm. Therefore, using matrices such that all rows $A_i$ have the same norm makes it simpler to obtain the quantum access.

The time or complexity $T$ necessary for the quantum access can be reduced to polylogarithmic dependence in $n$ and $d$ if we consider the access to a Quantum Memory or *QRAM*. The QRAM Kerenidis & Prakash (2017a) is a specific data structure from which a quantum circuit can allow quantum access to data in time $O(\log (nd))$.

**Theorem B.1 (QRAM data structure, see Kerenidis & Prakash (2017a))** *Let $A \in \mathbb{R}^{n \times d}$, there is a data structure to store the rows of $A$ such that,*

1. *The time to insert, update or delete a single entry $A_{ij}$ is $O(\log^2(n))$.*

2. *A quantum algorithm with access to the data structure can perform the following unitaries in time $T = O(\log^2 n)$.*

   *(a) $|i\rangle |0\rangle \to |i\rangle |A_i\rangle$ for $i \in [n]$.*
   *(b) $|0\rangle \to \sum_{i \in [n]} \|A_i\| |i\rangle$.*

We now state important methods for processing the quantum information. Their goal is to store some information alternatively in the quantum state's amplitude or in the quantum register as a bitstring.

**Theorem B.2** *[Amplitude Amplification and Estimation Brassard et al. (2002)] Given a unitary operator $U$ such that $U : |0\rangle \mapsto \sqrt{p} |y\rangle |0\rangle + \sqrt{1-p} |y^\perp\rangle |1\rangle$ in time $T$, where $p > 0$ is the probability of measuring "0", it is possible to obtain the state $|y\rangle |0\rangle$ using $O(\frac{T}{\sqrt{p}})$ queries to $U$, or to estimate $p$ with relative error $\delta$ using $O(\frac{T}{\delta \sqrt{p}})$ queries to $U$.*

**Theorem B.3** *[Conditional Rotation] Given the quantum state $|a\rangle$, with $a \in [-1, 1]$, it is possible to perform $|a\rangle |0\rangle \mapsto |a\rangle (a |0\rangle + \sqrt{1-a} |1\rangle)$ with complexity $\widetilde{O}(1)$.*

Using Theorem F.3 followed by Theorem F.2, it then possible to transform the state $\frac{1}{\sqrt{d}} \sum_{j=0}^{d-1} |x_j\rangle$ into $\frac{1}{\|x\|} \sum_{j=0}^{d-1} x_j |x_j\rangle$.

In addition to amplitude estimation, we will make use of a tool developed in Wiebe et al. (2014a) to boost the probability of getting a good estimate for the inner product required for the quantum convolution algorithm. In high level, we take multiple copies of the estimator from the amplitude estimation procedure, compute the median, and reverse the circuit to get rid of the garbage. Here we provide a theorem with respect to time and not query complexity.

**Theorem B.4 (Median Evaluation, see Wiebe et al. (2014a))** *Let $\mathcal{U}$ be a unitary operation that maps*

$$\mathcal{U} : |0^{\otimes n}\rangle \mapsto \sqrt{a} |x, 1\rangle + \sqrt{1-a} |G, 0\rangle$$

*for some $1/2 < a \leq 1$ in time $T$. Then there exists a quantum algorithm that, for any $\Delta > 0$ and for any $1/2 < a_0 \leq a$, produces a state $|\Psi\rangle$ such that $\| |\Psi\rangle - |0^{\otimes nL}\rangle |x\rangle \| \leq \sqrt{2\Delta}$ for some integer*

*L, in time*

$$2T \left\lceil \frac{\ln(1/\Delta)}{2 \left( |a_0| - \frac{1}{2} \right)^2} \right\rceil.$$

### B.3 QUANTUM SUBROUTINES FOR LINEAR ALGEBRA

In the recent years, as the field of quantum machine learning grew, its "toolkit" for linear algebra algorithms has become important enough to allow the development of many quantum machine learning algorithms. We introduce here the important subroutines for this work, without detailing the circuits or the algorithms.

**Definition 2** *For a matrix $A$, the parameter $\mu(A)$ is defined by $\mu(A) = \min_{p \in [0,1]} \left( \|A\|_F , \sqrt{s_{2p}(A)s_{2(1-p)}(A^T)} \right)$ where $s_p(A) = \max_i(\|A_i\|_p^p)$.*

The next theorems allow to compute the distance between vectors encoded as quantum states, and use this idea to perform the $k$-means algorithm.

**Theorem B.5** *[Quantum Distance Estimation Wiebe et al. (2014b); Kerenidis et al. (2019)] Given quantum access in time $T$ to two matrices $U$ and $V$ with rows $u_i$ and $v_j$ of dimension $d$, there is a quantum algorithm that, for any pair $(i,j)$, performs the following mapping $|i\rangle |j\rangle |0\rangle \mapsto |i\rangle |j\rangle |\overline{d^2(u_i, v_j)}\rangle$, estimating the euclidean distance between $u_i$ and $v_j$ with precision $|\overline{d^2(u_i, v_j)} - d^2(u_i, v_j)| \leq \epsilon$ for any $\epsilon > 0$. The algorithm has a running time given by $\widetilde{O}(T\eta/\epsilon)$, where $\eta = \max_{ij}(\|u_i\| \|v_j\|)$, assuming that $\min_i(\|u_i\|) = \min_i(\|v_i\|) = 1$.*

**Theorem B.6** *[Quantum k-means clustering Kerenidis et al. (2019)] Given quantum access in time $T$ to a dataset $V \in \mathbb{R}^{n \times d}$, there is a quantum algorithm that outputs with high probability $k$ centroids $c_1, \cdots, c_k$ that are consistent with the output of the k-means algorithm with noise $\delta > 0$, in time $\widetilde{O}(T \times (kd\frac{\eta(V)}{\delta^2}\kappa(V)(\mu(V) + k\frac{\eta(V)}{\delta}) + k^2\frac{\eta(V)^{1.5}}{\delta^2}\kappa(V)\mu(V)))$ per iteration.*

**Definition 3** *For a matrix $V \in \mathbb{R}^{n \times d}$, its parameter $\eta(V)$ is defined as as $\frac{\max_i(\|v_i\|^2)}{\min_i(\|v_i\|^2)}$, or as $\max_i(\|v_i\|^2)$ assuming $\min_i(\|v_i\|) = 1$.*

In theorem F.6, the other parameters in the running time can be interpreted as follows : $\delta$ is the precision in the estimation of the distances, but also in the estimation of the position of the centroids. $\kappa(V)$ is the condition number of $V$ and $\mu(V)$ is defined above (Definition 5). Finally, in the case of *well clusterable datasets*, which should be the case when we will apply $k$-means during spectral clustering, the running simplifies to $\widetilde{O}(T \times (k^2 d\frac{\eta(V)^{2.5}}{\delta^3} + k^{2.5}\frac{\eta(V)^2}{\delta^3}))$.

Note that the dependence in $n$ is hidden in the time $T$ to load the data. This dependence becomes polylogarithmic in $n$ if we assume access to a QRAM.

**Theorem B.7 (Quantum Matrix Operations, Chakraborty et al. (2018) )** *Let $M \in \mathbb{R}^{d \times d}$ and $x \in \mathbb{R}^d$. Let $\delta_1, \delta_2 > 0$. If $M$ is stored in appropriate QRAM data structures and the time to prepare $|x\rangle$ is $T_x$, then there exist quantum algorithms that with probability at least $1 - 1/poly(d)$ return*

1. *A state $|z\rangle$ such that $\||z\rangle - |Mx\rangle\|_2 \leq \delta_1$ in time $\widetilde{O}((\kappa(M)\mu(M) + T_x\kappa(M))\log(1/\delta_1))$. Note that this also implies $\||z\rangle - |Mx\rangle\|_\infty \leq \delta_1$*

2. *Norm estimate $z \in (1 \pm \delta_2)\|Mx\|_2$, with relative error $\delta_2$, in time $\widetilde{O}(T_x\frac{\kappa(M)\mu(M)}{\delta_2}\log(1/\delta_1))$.*

The linear algebra procedures above can also be applied to any rectangular matrix $V \in \mathbb{R}^{n \times d}$ by considering instead the symmetric matrix $\overline{V} = \begin{pmatrix} 0 & V \\ V^T & 0 \end{pmatrix}$.

## APPENDIX C    CLASSICAL CONVOLUTIONAL NEURAL NETWORK (CNN)

CNN is a specific type of neural network, designed in particular for image processing or time series. It uses the *Convolution Product* as a main procedure for each layer. We will focus on image processing with a tensor framework for all elements of the network. Our goal is to explicitly describe the CNN procedures in a form that can be translated in the context of quantum algorithms.

As a regular neural network, a CNN should learn how to classify any input, in our case images. The training consists of optimizing a series of parameters, learned on the inputs and their corresponding labels.

### C.1    TENSOR REPRESENTATION

Images, or more generally layers of the network, can be seen as tensors. A tensor is a generalization of a matrix to higher dimensions. For instance an image of height $H$ and width $W$ can be seen as a matrix in $\mathbb{R}^{H \times W}$, where every pixel is a greyscale value between 0 ans 255 (8 bit). However the three channels of color (RGB: Red Green Blue) must be taken into account, by stacking three times the matrix for each color. The whole image is then seen as a 3 dimensional tensor in $\mathbb{R}^{H \times W \times D}$ where $D$ is the number of channels. We will see that the Convolution Product in the CNN can be expressed between 3-tensors (input) and 4-tensors (convolution *filters* or *kernels*), the output being a 3-tensor of different dimensions (spatial size and number of channels).

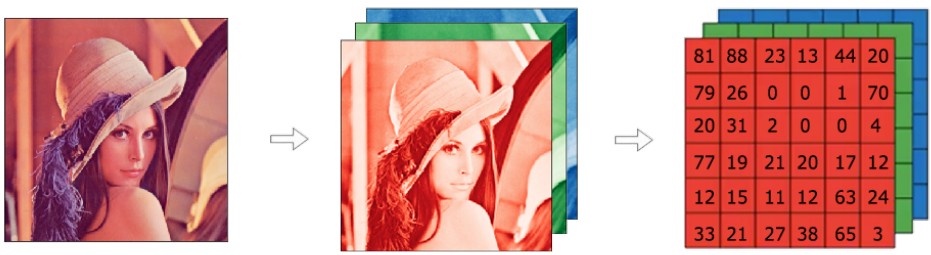

Figure 3: RGB decomposition, a colored image is a 3-tensor.

### C.2    ARCHITECTURE

A CNN is composed of 4 main procedures, compiled and repeated in any order : Convolution layers, most often followed by an Activation Function, Pooling Layers and some Fully Connected layers at the end. We will note $\ell$ the current layer.

**Convolution Layer :**    The $\ell^{th}$ layer is convolved by a set of filters called *kernels*. The output of this operation is the $(\ell + 1)^{th}$ layer. A convolution by a single kernel can be seen as a feature detector, that will screen over all regions of the input. If the feature represented by the kernel, for instance a vertical edge, is present in some part of the input, there will be a high value at the corresponding position of the output. The output is commonly called the *feature map* of this convolution.

**Activation Function :**    As in regular neural network, we insert some non linearities also called *activation functions*. These are mandatory for a neural network to be able to learn any function. In the case of a CNN, each convolution is often followed by a Rectified Linear Unit function, or *ReLu*. This is a simple function that puts all negative values of the output to zero, and lets the positive values as they are.

**Pooling Layer :**    This downsampling technique reduces the dimensionality of the layer, in order to improve the computation. Moreover, it gives to the CNN the ability to learn a representation invariant to small translations. Most of the time, we apply a Maximum Pooling or an Average Pooling. The first one consists of replacing a subregion of $P \times P$ elements only by the one with the

maximum value. The second does the same by averaging all values. Recall that the value of a pixel corresponds to how much a particular feature was present in the previous convolution layer.

**Fully Connected Layer :**   After a certain number of convolution layers, the input has been sufficiently processed so that we can apply a fully connected network. Weights connect each input to each output, where inputs are all element of the previous layer. The last layer should have one node per possible label. Each node value can be interpreted as the probability of the initial image to belong to the corresponding class.

### C.3   CONVOLUTION PRODUCT AS A TENSOR OPERATION

Most of the following mathematical formulations have been very well detailed by Wu (2017). At layer $\ell$, we consider the convolution of a multiple channels image, seen as a 3-tensor $X^\ell \in \mathbb{R}^{H^\ell \times W^\ell \times D^\ell}$. Let's consider a single kernel in $\mathbb{R}^{H \times W \times D^\ell}$. Note that its third dimension must match the number of channels of the input, as in Figure 4. The kernel passes over all possible regions of the input and outputs a value for each region, stored in the corresponding element of the output. Therefore the output is 2 dimensional, in $\mathbb{R}^{H^{\ell+1} \times W^{\ell+1}}$

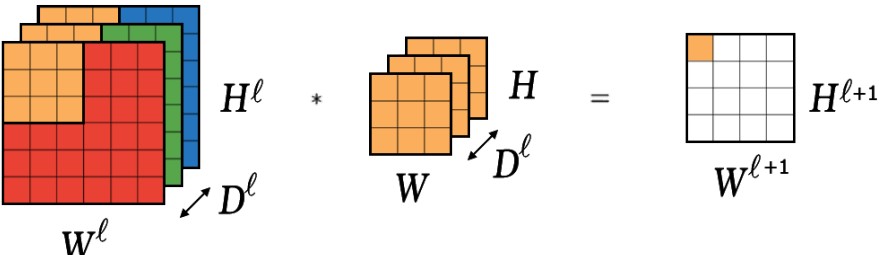

Figure 4: Convolution of a 3-tensor input (Left) by one 3-tensor kernel (Center). The ouput (Right) is a matrix for which each entry is a inner product between the kernel and the corresponding overlapping region of the input.

In a CNN, the most general case is to apply several convolution products to the input, each one with a different 3-tensor kernel. Let's consider an input convolved by $D^{\ell+1}$ kernels. We can globally see this process as a whole, represented by one 4-tensor kernel $K^\ell \in \mathbb{R}^{H \times W \times D^\ell \times D^{\ell+1}}$. As $D^{\ell+1}$ convolutions are applied, there are $D^{\ell+1}$ outputs of 2 dimensions, equivalent to a 3-tensor $X^{\ell+1} \in \mathbb{R}^{H^{\ell+1} \times W^{\ell+1} \times D^{\ell+1}}$

We can see on Figure 5 that the output's dimensions are modified given the following rule:

$$\begin{cases} H^{\ell+1} = H^\ell - H + 1 \\ W^{\ell+1} = W^\ell - W + 1 \end{cases} \tag{4}$$

We omit to detail the use of *Padding* and *Stride*, two parameters that control how the kernel moves through the input, but these can easily be incorporated in the algorithms.

An element of $X^\ell$ is determined by 3 indices $(i^\ell, j^\ell, d^\ell)$, while an element of the kernel $K^\ell$ is determined by 4 indices $(i, j, d, d')$. For an element of $X^{\ell+1}$ we use 3 indices $(i^{\ell+1}, j^{\ell+1}, d^{\ell+1})$. We can express the value of each element of the output $X^{\ell+1}$ with the relation

$$X^{\ell+1}_{i^{\ell+1}, j^{\ell+1}, d^{\ell+1}} = \sum_{i=0}^{H} \sum_{j=0}^{W} \sum_{d=0}^{D^\ell} K^\ell_{i,j,d,d^{\ell+1}} X^\ell_{i^{\ell+1}+i, j^{\ell+1}+j, d} \tag{5}$$

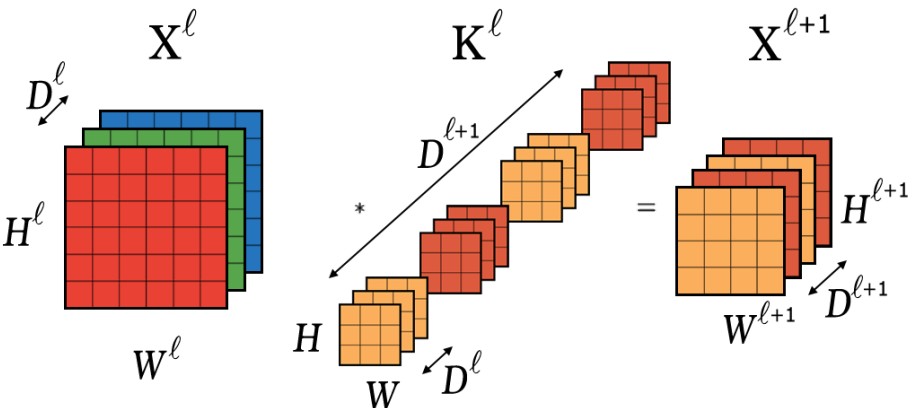

Figure 5: Convolutions of the 3-tensor input $X^\ell$ (Left) by one 4-tensor kernel $K^\ell$ (Center). Each channel of the output $X^{\ell+1}$ (Right) corresponds to the output matrix of the convolution with one of the 3-tensor kernel.

### C.4 MATRIX EXPRESSION

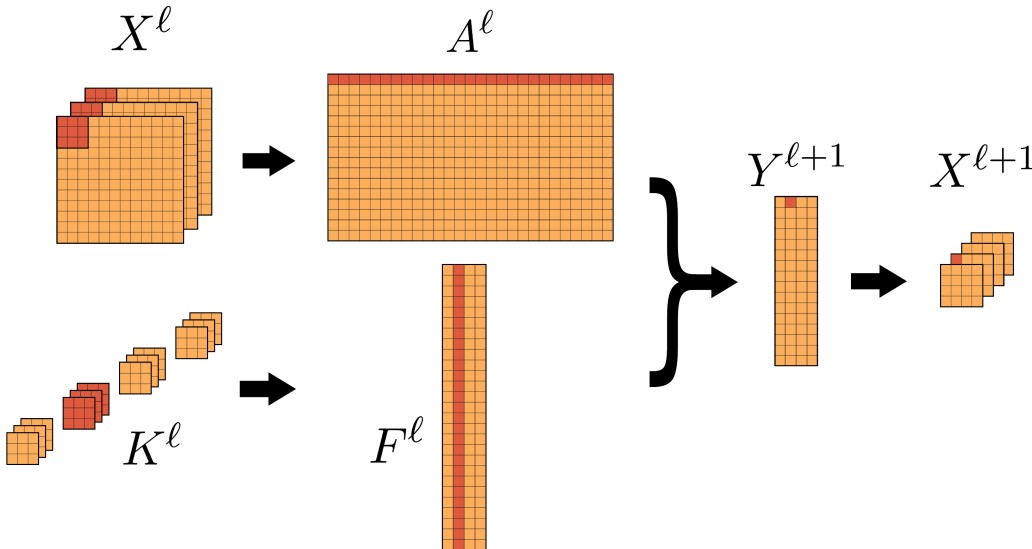

Figure 6: A convolution product is equivalent to a matrix-matrix multiplication.

It is possible to reformulate Equation (5) as a matrix product. For this we have to reshape our objects. We expand the input $X^\ell$ into a matrix $A^\ell \in \mathbb{R}^{(H^{\ell+1}W^{\ell+1})\times(HWD^\ell)}$. Each row of $A^\ell$ is a vectorized version of a subregion of $X^\ell$. This subregion is a volume of the same size as a single kernel volume $H \times W \times D^\ell$. Hence each of the $H^{\ell+1} \times W^{\ell+1}$ rows of $A^\ell$ is used for creating one value in $X^{\ell+1}$. Given such a subregion of $X^\ell$, the rule for creating the row of $A^\ell$ is to stack, channel by channel, a column first vectorized form of each matrix. Then, we reshape the kernel tensor $K^\ell$ into a matrix $F^\ell \in \mathbb{R}^{(HWD^\ell)\times D^{\ell+1}}$, such that each column of $F^\ell$ is a column first vectorized version of one of the $D^{\ell+1}$ kernels.

The convolution operation $X^\ell * K^\ell = X^{\ell+1}$ is equivalent to the following matrix multiplication

$$A^\ell F^\ell = Y^{\ell+1}, \tag{6}$$

where each column of $Y^{\ell+1} \in \mathbb{R}^{(H^{\ell+1}W^{\ell+1}) \times D^{\ell+1}}$ is a column first vectorized form of one of the $D^{\ell+1}$ channels of $X^{\ell+1}$. Note that an element $Y_{p,q}^{\ell+1}$ is the inner product between the $p^{th}$ row of $A^{\ell}$ and the $q^{th}$ column of $F^{\ell}$. It is then simple to convert $Y^{\ell+1}$ into $X^{\ell+1}$ The indices relation between the elements $Y_{p,q}^{\ell+1}$ and $X_{i^{\ell+1},j^{\ell+1},d^{\ell+1}}^{\ell+1}$ is given by:

$$\begin{cases} d^{\ell+1} = q \\ j^{\ell+1} = \lfloor \frac{p}{H^{\ell+1}} \rfloor \\ i^{\ell+1} = p - H^{\ell+1} \lfloor \frac{p}{H^{\ell+1}} \rfloor \end{cases} \tag{7}$$

A summary of all variables along with their meaning and dimensions is given in Section A.

## APPENDIX D    QUANTUM CONVOLUTIONAL NEURAL NETWORK

In this section we will design quantum procedures for the usual operations in a CNN layer. We start by describing the main ideas before providing the details. Steps are gathered in Algorithm 1.

First, to perform a convolution product between an input and a kernel, we will use the mapping between convolution of tensors and matrix multiplication from Section C.3, that can further be reduced to inner product estimation between vectors, in order to use quantum linear algebra procedures to perform these computations faster. The output will be a quantum state representing the result of the convolution product, from which we can sample to retrieve classical information to feed the next layer. This is stated by the following Theorem:

**Theorem D.1** *(Quantum Convolution Layer)*
*Given 3D tensor input $X^{\ell} \in \mathbb{R}^{H^{\ell} \times W^{\ell} \times D^{\ell}}$ and 4D tensor kernel $K^{\ell} \in \mathbb{R}^{H \times W \times D^{\ell} \times D^{\ell+1}}$ stored in QRAM, there is a quantum algorithm that computes a quantum states $\Delta$-close to $|f(\overline{X}^{\ell+1})\rangle$ with arbitrary small parameter $\Delta > 0$. $|f(\overline{X}^{\ell+1})\rangle$ is close to the result of the convolution product $X^{\ell+1} = X^{\ell} * K^{\ell}$ followed by any non linear function $f : \mathbb{R} \mapsto \mathbb{R}^{+}$, with an error bounded by $\left\| f(\overline{X}^{\ell+1}) - f(X^{\ell+1}) \right\|_{\infty} \leq 2M\epsilon$ for any precision $\epsilon > 0$, where $M$ is the maximum norm of a product between one of the $D^{\ell+1}$ kernels, and one of the regions of $X^{\ell}$ of size $HWD^{\ell}$. The time complexity of this procedure is given by $\widetilde{O}(1/\epsilon)$, where $\widetilde{O}$ hides factors poly-logarithmic in $\Delta$ and in the size of $X^{\ell}$ and $K^{\ell}$.*

In a second step, we efficiently retrieve classical information from the output. Recall that a convolution can be seen as a pattern detection on the input image, where the pattern is the kernel. The output values correspond to "how much" the pattern was present in the corresponding region of the input. Low value pixels in the output indicate the absence of the pattern in the input at the corresponding regions. Therefore, by sampling according to these output values, where the high value pixels are sampled with more probability, we could retrieve less but only meaningful information for the neural network to learn. While sampling, we update the QRAM data structure with the new information (see Section D.2). We also perform the Pooling operation during this phase (see Section D.2.2). It is an interesting use case where amplitudes of a quantum state are proportional to the importance of the information they carry, giving a new utility to the probabilistic nature of quantum sampling. Numerical simulations are presented in Section 6 to have an empirical estimation of how many samples from the output state are necessary.

### D.1    SINGLE QUANTUM CONVOLUTION LAYER

In order to develop a quantum algorithm to perform the convolution as described above, we will make use of quantum linear algebra procedures. We will use quantum states proportional to the rows of $A^{\ell}$, noted $|A_p\rangle$, and the columns of $F^{\ell}$, noted $|F_q\rangle$ (we omit the $\ell$ exponent in the quantum states to simplify the notation). These states are given by $|A_p\rangle = \frac{1}{\|A_p\|} \sum_{r=0}^{HWD^{\ell}-1} A_{pr} |r\rangle$ and $|F_q\rangle = \frac{1}{\|F_q\|} \sum_{s=0}^{D^{\ell+1}-1} F_{sq} |s\rangle$. We suppose we can load these vectors in quantum states by performing the following queries:

$$\begin{cases} |p\rangle |0\rangle \mapsto |p\rangle |A_p\rangle \\ |q\rangle |0\rangle \mapsto |q\rangle |F_q\rangle \end{cases} \tag{8}$$

Such queries, in time poly-logarithmic in the dimension of the vector, can be implemented with a Quantum Random Access Memory (QRAM). See Section D.2 for more details on the QRAM update rules and its integration layer by layer.

### D.1.1 INNER PRODUCT ESTIMATION

The following method to estimates inner product is derived from previous work by Kerenidis et al. (2019). With the initial state $|p\rangle |q\rangle \frac{1}{\sqrt{2}}(|0\rangle + |1\rangle)|0\rangle$ we apply the queries detailed above in a controlled fashion, followed simply by a Hadamard gate to extract the inner product $\langle A_p | F_q \rangle$ in an amplitude. $\frac{1}{\sqrt{2}}(|p\rangle |q\rangle |0\rangle |0\rangle + |p\rangle |q\rangle |1\rangle |0\rangle) \mapsto \frac{1}{\sqrt{2}}(|p\rangle |q\rangle |0\rangle |A_p\rangle + |p\rangle |q\rangle |1\rangle |F_q\rangle)$. By applying a Hadamard gate on the third register we obtain the following state, $\frac{1}{2}|p\rangle |q\rangle \Big(|0\rangle(|A_p\rangle + |F_q\rangle) + |1\rangle(|A_p\rangle - |F_q\rangle)\Big)$. The probability of measuring $0$ on the third register is given by $P_{pq} = \frac{1+\langle A_p|F_q\rangle}{2}$. Thus we can rewrite the previous state as $|p\rangle |q\rangle \Big(\sqrt{P_{pq}}|0, y_{pq}\rangle + \sqrt{1-P_{pq}}|1, y_{pq}'\rangle\Big)$, where $|y_{pq}\rangle$ and $|y_{pq}'\rangle$ are some garbage states. We can perform the previous circuit in superposition. Since $A^\ell$ has $H^{\ell+1}W^{\ell+1}$ rows, and $F^\ell$ has $D^{\ell+1}$ columns, we obtain the state: $|u\rangle = \frac{1}{\sqrt{H^{\ell+1}W^{\ell+1}D^{\ell+1}}} \sum_p \sum_q |p\rangle |q\rangle \Big(\sqrt{P_{pq}}|0, y_{pq}\rangle + \sqrt{1-P_{pq}}|1, y_{pq}'\rangle\Big)$ Therefore the probability of measuring the triplet $(p, q, 0)$ in the first three registers is given by $P_0(p, q) = \frac{P_{pq}}{H^{\ell+1}W^{\ell+1}D^{\ell+1}} = \frac{1+\langle A_p|F_q\rangle}{2H^{\ell+1}W^{\ell+1}D^{\ell+1}}$ Now we can relate to the Convolution product. Indeed, the triplets $(p, q, 0)$ that are the most probable to be measured are the ones for which the value $\langle A_p|F_q\rangle$ is the highest. Recall that each element of $Y^{\ell+1}$ is given by $Y_{pq}^{\ell+1} = (A_p, F_q)$, where "$(\cdot, \cdot)$" denotes the inner product. We see here that we will sample most probably the positions $(p, q)$ for the highest values of $Y^{\ell+1}$, that corresponds to the most important points of $X^{\ell+1}$, by the Equation (7). Note that the the values of $Y^{\ell+1}$ can be either positive of negative, which is not an issue thanks to the positiveness of $P_0(p, q)$.

A first approach could be to measure indices $(p, q)$ and rely on the fact that pixels with high values, hence a high amplitude, would have a higher probability to be measured. However we have not exactly the final result, since $\langle A_p|F_q\rangle \neq (A_p, F_q) = \|A_p\| \|F_q\| \langle A_p|F_q\rangle$. Most importantly we then want to apply a non linearity $f(Y_{pq}^{\ell+1})$ to each pixel, for instance the ReLu function, which seems not possible with unitary quantum gates if the data is encoded in the amplitudes only. Morever, due to normalization of the quantum amplitudes and the high dimension of the Hilbert space of the input, the probability of measuring each pixel is roughly the same, making the sampling inefficient. Given these facts, we have added steps to the circuit, in order to measure $(p, q, f(Y_{pq}^{\ell+1}))$, therefore know the value of a pixel when measuring it, while still measuring the most important points in priority.

### D.1.2 ENCODING THE AMPLITUDE IN A REGISTER

Let $\mathcal{U}$ be the unitary that map $|0\rangle$ to $|u\rangle$: $|u\rangle = \frac{1}{\sqrt{H^{\ell+1}W^{\ell+1}D^{\ell+1}}} \sum_{p,q} |p\rangle |q\rangle \Big(\sqrt{P_{pq}}|0, y_{pq}\rangle + \sqrt{1-P_{pq}}|1, y_{pq}'\rangle\Big)$. The amplitude $\sqrt{P_{pq}}$ can be encoded in an ancillary register by using Amplitude Estimation (Theorem F.2) followed by a Median Evaluation (Theorem F.4). For any $\Delta > 0$ and $\epsilon > 0$, we can have a state $\Delta$-close to $|u'\rangle = \frac{1}{\sqrt{H^{\ell+1}W^{\ell+1}D^{\ell+1}}} \sum_{p,q} |p\rangle |q\rangle |0\rangle |\overline{P}_{pq}\rangle |g_{pq}\rangle$ with probability at least $1 - 2\Delta$, where $|P_{pq} - \overline{P}_{pq}| \leq \epsilon$ and $|g_{pq}\rangle$ is a garbage state. This requires $O(\frac{\ln(1/\Delta)}{\epsilon})$ queries of $\mathcal{U}$. In the following we discard the third register $|0\rangle$ for simplicity.

The benefit of having $\overline{P}_{pq}$ in a register is to be able to perform operations on it (arithmetic or even non linear). Therefore we can simply obtain a state corresponding to the exact value of the the convolution product. Since we've built a circuit such that $P_{pq} = \frac{1+\langle A_p|F_q\rangle}{2}$, with two QRAM calls, we can retrieve the norm of the vectors by applying the following unitary $|p\rangle |q\rangle |\overline{P}_{pq}\rangle |g_{pq}\rangle |0\rangle |0\rangle \mapsto |p\rangle |q\rangle |\overline{P}_{pq}\rangle |g_{pq}\rangle |\|A_p\|\rangle |\|F_q\|\rangle$. On the fourth register, we can then write $Y_{pq}^{\ell+1} = \|A_p\| \|F_q\| \langle A_p|F_q\rangle$ using some arithmetic circuits (addition, multiplication by a scalar, multiplication between registers). We then apply a boolean circuit that implements the ReLu function on the same register, in order to obtain an estimate of $f(Y_{pq}^{\ell+1})$ in the fourth register. We finish by inverting the previous computations and obtain the final state:

$$|f(\overline{Y}^{\ell+1})\rangle = \frac{1}{\sqrt{H^{\ell+1}W^{\ell+1}D^{\ell+1}}} \sum_{p,q} |p\rangle |q\rangle |f(\overline{Y}_{pq}^{\ell+1})\rangle |g_{pq}\rangle \qquad (9)$$

Because of the precision $\epsilon$ on $|\overline{P}_{pq}\rangle$, our estimation $\overline{Y}_{pq}^{\ell+1} = (2\overline{P}_{pq} - 1) \|A_p\| \|F_q\|$, is obtained with error such that $|\overline{Y}_{pq}^{\ell+1} - Y_{pq}^{\ell+1}| \leq 2\epsilon \|A_p\| \|F_q\|$.

In superposition, we can bound this error by $|\overline{Y}_{pq}^{\ell+1} - Y_{pq}^{\ell+1}| \leq 2M\epsilon$ where we define

$$M = \max_{p,q} \|A_p\| \|F_q\| \qquad (10)$$

$M$ is the maximum product between norms of one of the $D^{\ell+1}$ kernels, and one of the regions of $X^\ell$ of size $HWD^\ell$. Finally, since the previous error estimation is valid for all pairs $(p,q)$, the overall error committed on the convolution product can be bounded by $\left\|\overline{Y}^{\ell+1} - Y^{\ell+1}\right\|_\infty \leq 2M\epsilon$, where $\|.\|_\infty$ denotes the $\ell_\infty$ norm. Recall that $Y^{\ell+1}$ is just a reshaped version of $X^{\ell+1}$. Since the non linearity adds no approximation, we can conclude on the final error committed for a layer of our QCNN

$$\left\|f(\overline{X}^{\ell+1}) - f(X^{\ell+1})\right\|_\infty \leq 2M\epsilon \qquad (11)$$

At this point, we have established Theorem D.1 as we have created the quantum state (9), with given precision guarantees, in time poly-logarithmic in $\Delta$ and in the size of $X^\ell$ and $K^\ell$.

We know aim to retrieve classical information from this quantum state. Note that $|Y_{pq}^{\ell+1}\rangle$ is representing a scalar encoded in as many qubits as needed for the precision, whereas $|A_p\rangle$ was representing a vector as a quantum state in superposition, where each element $A_{p,r}$ is encoded in one amplitude (See Section F). The next step can be seen as a way to retrieve both encoding at the same time, that will allow an efficient tomography focus on the values of high magnitude.

### D.1.3 CONDITIONAL ROTATION

In the following sections, we omit the $\ell + 1$ exponent for simplicity. Garbage states are removed as they will not perturb the final measurement. We now aim to modify the amplitudes, such that the highest values of $|f(\overline{Y})\rangle$ are measured with higher probability. A way to do so consists in applying a conditional rotation on an ancillary qubit, proportionally to $f(\overline{Y}_{pq})$. We will detail the calculation since in the general case $f(\overline{Y}_{pq})$ can be greater than 1. To simplify the notation, we note $x = f(\overline{Y}_{pq})$. This step consists in applying the following rotation on a ancillary qubit: $|x\rangle |0\rangle \mapsto |x\rangle \left(\sqrt{\frac{x}{\max x}} |0\rangle + \beta |1\rangle\right)$, where $\max x = \max_{p,q} f(\overline{Y}_{pq})$ and $\beta = \sqrt{1 - (\frac{x}{\max x})^2}$. Note that in practice it is not possible to have access to $|\max x\rangle$ from the state (9), but we will present a method to know *a priori* this value or an upper bound in section D.1.6. Let's note $\alpha_{pq} = \sqrt{\frac{f(\overline{Y}_{pq})}{max_{p,q}(f(\overline{Y}_{pq}))}}$. The ouput of this conditional rotation in superposition on state (9) is then $\frac{1}{\sqrt{HWD}} \sum_{p,q} |p\rangle |q\rangle |f(\overline{Y}_{pq})\rangle (\alpha_{pq} |0\rangle + \sqrt{1 - \alpha_{pq}^2} |1\rangle)$.

### D.1.4 AMPLITUDE AMPLIFICATION

In order to measure $(p, q, f(\overline{Y}_{pq}))$ with higher probability where $f(\overline{Y}_{pq})$ has high value, we could post select on the measurement of $|0\rangle$ on the last register. Otherwise, we can perform an amplitude amplification on this ancillary qubit. Let's rewrite the previous state as $\frac{1}{\sqrt{HWD}} \sum_{p,q} \alpha_{pq} |p\rangle |q\rangle |f(\overline{Y}_{pq})\rangle |0\rangle + \sqrt{1 - \alpha_{pq}^2} |g'_{pq}\rangle |1\rangle$, where $|g'_{pq}\rangle$ is another garbage state. The overall probability of measuring $|0\rangle$ on the last register is $P(0) = \frac{1}{HWD} \sum_{pq} |\alpha_{pq}|^2$. The number of queries required to amplify the state $|0\rangle$ is $O(\frac{1}{\sqrt{P(0)}})$, as shown by Brassard et al. (2002). Since $f(\overline{Y}_{pq}) \in \mathbb{R}^+$, we have $\alpha_{pq}^2 = \frac{f(\overline{Y}_{pq})}{\max_{p,q}(f(\overline{Y}_{pq}))}$. Therefore the number of

queries is $O\left(\sqrt{\max_{p,q}(f(\overline{Y}_{pq}))} \frac{1}{\sqrt{\frac{1}{HWD}\sum_{p,q}f(\overline{Y}_{pq})}}\right) = O\left(\frac{\sqrt{\max_{p,q}(f(\overline{Y}_{pq}))}}{\sqrt{\mathbb{E}_{p,q}(f(\overline{Y}_{pq}))}}\right)$, where the nota-

tion $\mathbb{E}_{p,q}(f(\overline{Y}_{pq}))$ represents the average value of the matrix $f(\overline{Y})$. It can also be written $\mathbb{E}(f(\overline{X}))$ as in Result 1: $\mathbb{E}_{p,q}(f(\overline{Y}_{pq})) = \frac{1}{HWD}\sum_{p,q}f(\overline{Y}_{pq})$. At the end of these iterations, we have modified with high probability the state to the following:

$$|f(\overline{Y})\rangle = \frac{1}{\sqrt{HWD}}\sum_{p,q}\alpha'_{pq}|p\rangle|q\rangle|f(\overline{Y}_{pq})\rangle \tag{12}$$

Where, to respect the normalization of the quantum state, $\alpha'_{pq} = \frac{\alpha_{pq}}{\sqrt{\sum_{p,q}\frac{\alpha^2_{pq}}{HWD}}}$. Eventually, the

probability of measuring $(p, q, f(\overline{Y}_{pq}))$ is given by $p(p, q, f(\overline{Y}_{pq})) = \frac{(\alpha'_{pq})^2}{HWD} = \frac{f(\overline{Y}_{pq})}{\sum_{p,q}f(\overline{Y}_{pq})}$. Note

that we have used the same type of name $|f(\overline{Y})\rangle$ for both state (9) and state (12). For now on, this state name will refer only to the latter (12).

### D.1.5   $\ell_\infty$ TOMOGRAPHY AND PROBABILISTIC SAMPLING

We can rewrite the final quantum state obtained in (12) as

$$|f(\overline{Y}^{\ell+1})\rangle = \frac{1}{\sqrt{\sum_{p,q}f(\overline{Y}^{\ell+1}_{pq})}}\sum_{p,q}\sqrt{f(\overline{Y}^{\ell+1}_{pq})}|p\rangle|q\rangle|f(\overline{Y}^{\ell+1}_{pq})\rangle \tag{13}$$

We see here that $f(\overline{Y}^{\ell+1}_{pq})$, the values of each pixel, are encoded in both the last register and in the amplitude. We will use this property to extract efficiently the exact values of high magnitude pixels. For simplicity, we will use instead the notation $f(\overline{X}^{\ell+1}_n)$ to denote a pixel's value, with $n \in [H^{\ell+1}W^{\ell+1}D^{\ell+1}]$. Recall that $Y^{\ell+1}$ and $X^{\ell+1}$ are reshaped version of the same object.

The pixels with high values will have more probability of being sampled. Specifically, we perform a tomography with $\ell_\infty$ guarantee and precision parameter $\eta > 0$. See Theorem G.1 and Section G for details. The $\ell_\infty$ guarantee allows to obtain each pixel with error at most $\eta$, and require $\widetilde{O}(1/\eta^2)$ samples from the state (13). Pixels with low values $f(\overline{X}^{\ell+1}_n) < \eta$ will probably not be sampled due to their low amplitude. Therefore the error committed will be significative and we adopt the rule of setting them to 0. Pixels with higher values $f(\overline{X}^{\ell+1}_n) \geq \eta$, will be sample with high probability, and only one appearance is enough to get the exact register value $f(\overline{X}^{\ell+1}_n)$ of the pixel, as is it also written in the last register.

To conclude, let's note $\mathcal{X}^{\ell+1}_n$ the resulting pixel values after the tomography, and compare it to the real classical outputs $f(X^{\ell+1}_n)$. Recall that the measured values $f(\overline{X}^{\ell+1}_n)$ are approximated with error at most $2M\epsilon$ with $M = \max_{p,q}\|A_p\|\|F_q\|$. The algorithm described above implements the following rules:

$$\begin{cases} |\mathcal{X}^{\ell+1}_n - f(X^{\ell+1}_n)| \leq 2M\epsilon & \text{if} \quad f(\overline{X}^{\ell+1}_n) \geq \eta \\ \mathcal{X}^{\ell+1}_n = 0 & \text{if} \quad f(\overline{X}^{\ell+1}_n) < \eta \end{cases} \tag{14}$$

Concerning the running time, one could ask what values of $\eta$ are sufficient to obtain enough meaningful pixels. Obviously this highly depends on the output's size $H^{\ell+1}W^{\ell+1}D^{\ell+1}$ and on the output's content itself. But we can view this question from an other perspective, by considering that we sample a constant fraction of pixels given by $\sigma \cdot (H^{\ell+1}W^{\ell+1}D^{\ell+1})$ where $\sigma \in [0, 1]$ is a sampling ratio. Because of the particular amplitudes of state (13), the high value pixels will be measured and known with higher probability. The points that are not sampled are being set to 0. We see that this approach is equivalent to the $\ell_\infty$ tomography, therefore we have $\frac{1}{\eta^2} = \sigma \cdot H^{\ell+1}W^{\ell+1}D^{\ell+1}$.

We will use this analogy in the numerical simulations (Section 6) to estimate, for a particular QCNN architecture and a particular dataset of images, which values of $\sigma$ are enough to allow the neural network to learn.

### D.1.6 REGULARIZATION OF THE NON LINEARITY

In the previous steps, we see several appearances of the parameter $\max_{p,q}(f(\overline{Y}_{pq}^{\ell+1}))$. First for the conditional rotation preprocessing, we need to know this value or an upper bound. Then for the running time, we would like to bound this parameter. Both problems can be solved by replacing the usual ReLu non linearity by a particular activation function, that we note $capReLu$. This function is simply a parametrized ReLu function with an upper threshold, the cap $C$, after which the function remain constant. The choice of $C$ will be tuned for each particular QCNN, as a tradeoff between accuracy and speed. Otherwise, the only other requirement of the QCNN activation function would be not to allow negative values. This is already often the case for most of the classical CNN. In practice, we expect the capReLu to be as good as a usual ReLu, for convenient values of the cap $C$ ($\leq 10$). We performed numerical simulations to compare the learning curve of the same CNN with several values of $C$. See the numerical experiments presented in Section 6 for more details.

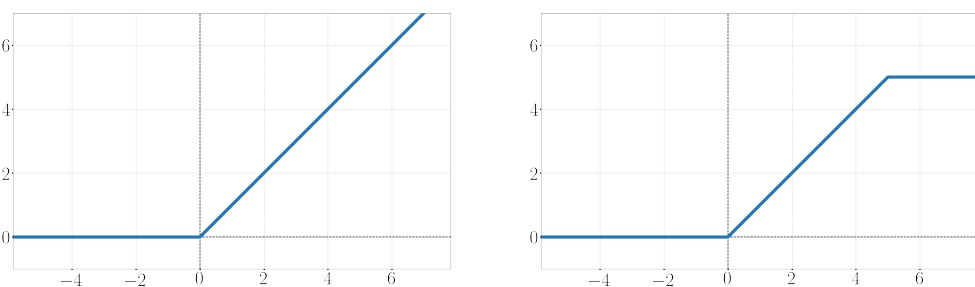

Figure 7: Activation functions: ReLu (Left) and capReLu (Right) with a cap $C$ at 5.

### D.2 QRAM UPDATE

We wish to detail the use of the QRAM between each quantum convolution layer, and present how the pooling operation can happen during this phase. General results about the QRAM is given as Theorem F.1. Implementation details can be found in the work of Kerenidis & Prakash (2017a). In this section, we will show how to store samples from the output of the layer $\ell$, to create the input of layer $\ell + 1$.

### D.2.1 STORING THE OUTPUT VALUES DURING THE SAMPLING

At the beginning of layer $\ell+1$, the QRAM must store $A^{\ell+1}$, a matrix where each elements is indexed by $(p', r')$, and perform $|p'\rangle |0\rangle \mapsto |p'\rangle |A_{p'}^{\ell+1}\rangle$. The data is stored in the QRAM as a tree structure described by Kerenidis & Prakash (2017b). Each row $A_{p'}^{\ell+1}$ is stored in such a tree $T_{p'}^{\ell+1}$. Each leaf $A_{p'r'}^{\ell+1}$ correspond to a value sampled from the previous quantum state $|f(\overline{Y}^{\ell+1})\rangle$, output of the layer $\ell$. The question is to know where to store a sample from $|f(\overline{Y}^{\ell+1})\rangle$ in the tree $T_{p'}^{\ell+1}$.

When a point is sampled from the final state of the quantum convolution, at layer $\ell$, as described in Section D.1.4, we obtain a triplet corresponding to the two positions and the value of a point in the matrix $f(\overline{Y}^{\ell+1})$. We can know where this point belong in the input of layer $\ell + 1$, the tensor $X^{\ell+1}$, by Equation (7), since $Y^{\ell}$ is a reshaped version of $X^{\ell}$.

The position in $X^{\ell+1}$, noted $(i^{\ell+1}, j^{\ell+1}, d^{\ell+1})$, is then matched to several positions $(p', r')$ in $A^{\ell+1}$. For each $p'$, we write in the tree $T_{p'}^{\ell+1}$ the sampled value at leaf $r'$ and update its parent nodes, as required in the work of Kerenidis & Prakash (2017b). Note that leaves that weren't updated will be considered as zeros, corresponding to pixels with too low values, or not selected during pooling (see next section).

Having stored pixels in this way, we can then query $|p'\rangle|0\rangle \mapsto |p'\rangle|A^{\ell}_{p'}\rangle$, using the quantum circuit developed by Kerenidis & Prakash (2017b), where we correctly have $|A^{\ell+1}_{p'}\rangle = \frac{1}{\|A^{\ell+1}_{p'}\|}\sum_{r'}A^{\ell+1}_{p'r'}|r'\rangle$. Note that each tree has a logarithmic depth in the number of leaves, hence the running time of writing the output of the quantum convolution layer in the QRAM gives a marginal multiplicative increase, poly-logarithmic in the number of points sampled from $|f(\overline{Y}^{\ell+1})\rangle$, namely $O(\log(1/\eta^2))$.

### D.2.2 QUANTUM POOLING

As for the classical CNN, a QCNN should be able to perform pooling operations. We first detail the notations for classical pooling. At the end of layer $\ell$, we wish to apply a pooling operation of size P on the output $f(X^{\ell+1})$. We note $\tilde{X}^{\ell+1}$ the tensor after the pooling operation. For a point in $f(X^{\ell+1})$ at position $(i^{\ell+1}, j^{\ell+1}, d^{\ell+1})$, we know to which *pooling region* it belongs, corresponding to a position $(\tilde{i}^{\ell+1}, \tilde{j}^{\ell+1}, \tilde{d}^{\ell+1})$ in $\tilde{X}^{\ell+1}$:

$$\begin{cases} \tilde{d}^{\ell+1} = d^{\ell+1} \\ \tilde{j}^{\ell+1} = \lfloor \frac{j^{\ell+1}}{P} \rfloor \\ \tilde{i}^{\ell+1} = \lfloor \frac{i^{\ell+1}}{P} \rfloor \end{cases} \tag{15}$$

Figure 8: A $2\times2$ tensor pooling. A point in $f(X^{\ell+1})$ (left) is given by its position $(i^{\ell+1}, j^{\ell+1}, d^{\ell+1})$. A point in $\tilde{X}^{\ell+1}$ (right) is given by its position $(\tilde{i}^{\ell+1}, \tilde{j}^{\ell+1}, \tilde{d}^{\ell+1})$. Different *pooling regions* in $f(X^{\ell+1})$ have separate colours, and each one corresponds to a unique point in $\tilde{X}^{\ell+1}$.

We now show how any kind of pooling can be efficiently integrated to our QCNN structure. Indeed the pooling operation will occur during the QRAM update described above, at the end of a convolution layer. At this moment we will store sampled values according to the pooling rules.

In the quantum setting, the output of layer $\ell$ after tomography is noted $\mathcal{X}^{\ell+1}$. After pooling, we will describe it by $\tilde{\mathcal{X}}^{\ell+1}$, which has dimensions $\frac{H^{\ell+1}}{P} \times \frac{W^{\ell+1}}{P} \times D^{\ell+1}$. $\tilde{\mathcal{X}}^{\ell+1}$ will be effectively used as input for layer $\ell+1$ and its values should be stored in the QRAM to form the trees $\tilde{T}^{\ell+1}_{p'}$, related to the matrix expansion $\tilde{A}^{\ell+1}$.

However $\mathcal{X}^{\ell+1}$ is not known before the tomography is over. Therefore we have to modify the update rule of the QRAM to implement the pooling in an online fashion, each time a sample from $|f(\overline{X}^{\ell+1})\rangle$ is drawn. Since several sampled values of $|f(\overline{X}^{\ell+1})\rangle$ can correspond to the same leaf $\tilde{A}^{\ell+1}_{p'r'}$ (points in the same *pooling region*), we need an overwrite rule, that will depend on the type of pooling. In the case of Maximum Pooling, we simply update the leaf and the parent nodes if the new sampled value is higher that the one already written. In the case of Average Polling, we replace the actual value by the new averaged value.

In the end, any pooling can be included in the already existing QRAM update. In the worst case, the running time is increased by $\widetilde{O}(P/\eta^2)$, an overhead corresponding to the number of times we need to overwrite existing leaves, with $P$ being a small constant in most cases.

As we will see in Section E, the final positions $(p, q)$ that were sampled from $|f(\overline{X}^{\ell+1})\rangle$ and selected after pooling must be stored for further use during the backpropagation phase.

## D.3    RUNNING TIME

We will now summarise the running time for one forward pass of convolution layer $\ell$. With $\tilde{O}$ we hide the polylogaryhtmic factors. We first write the running time of the classical CNN layer, which is given by $\widetilde{O}\left(H^{\ell+1}W^{\ell+1}D^{\ell+1} \cdot HWD^{\ell}\right)$. For the QCNN, the previous steps prove Result 1 and can be implemented in time $\widetilde{O}\left(\frac{1}{\epsilon\eta^2} \cdot \frac{M\sqrt{C}}{\sqrt{\mathbb{E}(f(\overline{X}^{\ell+1}))}}\right)$. Note that, as explain in Section D.1.5, the quantum running time can also be written $\widetilde{O}\left(\sigma H^{\ell+1}W^{\ell+1}D^{\ell+1} \cdot \frac{M\sqrt{C}}{\epsilon\sqrt{\mathbb{E}(f(\overline{X}^{\ell+1}))}}\right)$, with $\sigma \in [0, 1]$ being the fraction of sampled elements among $H^{\ell+1}W^{\ell+1}D^{\ell+1}$ of them.

It is interesting to notice that the one quantum convolution layer can also include the ReLu operation and the Pooling operation in the same circuit, for no significant increase in the running time, whereas in the classical CNN each operation must be done on the whole data again.

## APPENDIX E    QUANTUM BACKPROGATION

The entire QCNN is made of multiple layers. For the last layer's output, we expect only one possible outcome, or a few in the case of a classification task, which means that the dimension of the quantum output is very small. A full tomography can be performed on the last layer's output in order to calculate the outcome. The loss $\mathcal{L}$ is then calculated, as a measure of correctness of the predictions compared to the ground truth. As the classical CNN, our QCNN should be able to perform the optimization of its weights (elements of the kernels) to minimize the loss by an iterative method.

**Theorem E.1**  *(Quantum Backpropagation for Quantum CNN)*
*Given the forward pass quantum algorithm in Algorithm 1, the input matrix $A^{\ell}$ and the kernel matrix $F^{\ell}$ stored in the QRAM for each layer $\ell$, and a loss function $\mathcal{L}$, there is a quantum backpropagation algorithm that estimates, for any precision $\delta > 0$, the gradient tensor $\frac{\partial\mathcal{L}}{\partial F^{\ell}}$ and update each element to perform gradient descent such that $\forall(s, q), \left|\frac{\partial\mathcal{L}}{\partial F^{\ell}_{s,q}} - \overline{\frac{\partial\mathcal{L}}{\partial F^{\ell}_{s,q}}}\right| \leq 2\delta\left\|\frac{\partial\mathcal{L}}{\partial F^{\ell}}\right\|_2$. Let $\frac{\partial\mathcal{L}}{\partial Y^{\ell}}$ be the gradient with respect to the $\ell^{th}$ layer. The running time of a single layer $\ell$ for quantum backpropagation is given by*

$$O\left(\left(\left(\mu(A^{\ell}) + \mu(\frac{\partial\mathcal{L}}{\partial Y^{\ell+1}})\right)\kappa(\frac{\partial\mathcal{L}}{\partial F^{\ell}}) + \left(\mu(\frac{\partial\mathcal{L}}{\partial Y^{\ell+1}}) + \mu(F^{\ell})\right)\kappa(\frac{\partial\mathcal{L}}{\partial Y^{\ell}})\right)\frac{\log 1/\delta}{\delta^2}\right) \quad (16)$$

*where for a matrix $V$, $\kappa(V)$ is the condition number and $\mu(V)$ is defined in Equation (5).*

### E.1    CLASSICAL BACKPROPAGATION

After each forward pass, the outcome is compared to the true labels and define a loss. We can update our weights by gradient descent to minimize this loss, and iterate. The main idea behind the backpropagation is to compute the derivatives of the loss $\mathcal{L}$, layer by layer, starting from the last one.

At layer $\ell$, the derivatives needed to perform the gradient descent are $\frac{\partial\mathcal{L}}{\partial F^{\ell}}$ and $\frac{\partial\mathcal{L}}{\partial Y^{\ell}}$. The first one represents the gradient of the final loss $\mathcal{L}$ with respect to each kernel element, a matrix of values that we will use to update the kernel weights $F^{\ell}_{s,q}$. The second one is the gradient of $\mathcal{L}$ with respect to the layer itself and is only needed to calculate the gradient $\frac{\partial\mathcal{L}}{\partial F^{\ell-1}}$ at layer $\ell - 1$.

### E.1.1    CONVOLUTION PRODUCT

We first consider a classical convolution layer without non linearity or pooling. Thus the output of layer $\ell$ is the same tensor as the input of layer $\ell + 1$, namely $X^{\ell+1}$ or equivalently $Y^{\ell+1}$. Assuming we know $\frac{\partial\mathcal{L}}{\partial X^{\ell+1}}$ or equivalently $\frac{\partial\mathcal{L}}{\partial Y^{\ell+1}}$, both corresponding to the derivatives of the $(\ell+1)^{th}$ layer's

input, we will show how to calculate $\frac{\partial \mathcal{L}}{\partial F^\ell}$, the matrix of derivatives with respect to the elements of the previous kernel matrix $F^\ell$. This is the main goal in order to optimize the kernel's weights.

The details of the following calculations can be found in the work of Wu (2017). We will use the notation $vec(X)$ to represents the vectorized form of any tensor $X$.

Recall that $A^\ell$ is the matrix expansion of the tensor $X^\ell$, whereas $Y^\ell$ is a matrix reshaping of $X^\ell$. By applying the chain rule $\frac{\partial \mathcal{L}}{\partial vec(F^\ell)^T} = \frac{\partial \mathcal{L}}{\partial vec(X^{\ell+1})^T} \frac{\partial vec(X^{\ell+1})}{\partial vec(F^\ell)^T}$, we can obtain:

$$\frac{\partial \mathcal{L}}{\partial F^\ell} = (A^\ell)^T \frac{\partial L}{\partial Y^{\ell+1}} \qquad (17)$$

See calculations details in the work of Wu (2017). Equation (17) shows that, to obtain the desired gradient, we can just perform a matrix-matrix multiplication between the transposed layer itself ($A^\ell$) and the gradient with respect to the previous layer ($\frac{\partial L}{\partial Y^{\ell+1}}$).

Equation (17) explains also why we will need to calculate $\frac{\partial \mathcal{L}}{\partial Y^\ell}$ in order to backpropagate through layer $\ell - 1$. To calculate it, we use the chain rule again for $\frac{\partial \mathcal{L}}{\partial vec(X^\ell)^T} = \frac{\partial \mathcal{L}}{\partial vec(X^{\ell+1})^T} \frac{\partial vec(X^{\ell+1})}{\partial vec(X^\ell)^T}$. Recall that a point in $A^\ell$, indexed by the pair $(p, r)$, can correspond to several triplets $(i^\ell, j^\ell, d^\ell)$ in $X^\ell$. We will use the notation $(p, r) \leftrightarrow (i^\ell, j^\ell, d^\ell)$ to express formally this relation. One can show that $\frac{\partial \mathcal{L}}{\partial Y^{\ell+1}}(F^\ell)^T$ is a matrix of same shape as $A^\ell$, and that the chain rule leads to a simple relation to calculate $\frac{\partial \mathcal{L}}{\partial Y^\ell}$ :

$$\left[\frac{\partial \mathcal{L}}{\partial X^\ell}\right]_{i^\ell, j^\ell, d^\ell} = \sum_{(p,r) \leftrightarrow (i^\ell, j^\ell, d^\ell)} \left[\frac{\partial \mathcal{L}}{\partial Y^{\ell+1}}(F^\ell)^T\right]_{p, r} \qquad (18)$$

We have shown how to obtain the gradients with respect to the kernels $F^\ell$ and to the layer itself $Y^\ell$ (or equivalently $X^\ell$).

### E.1.2  NON LINEARITY

The activation function has also an impact on the gradient. In the case of the ReLu, we should only cancel gradient for points with negative values. For points with positive value, the derivatives remain the same since the function is the identity. A formal relation can be given by

$$\left[\frac{\partial \mathcal{L}}{\partial X^{\ell+1}}\right]_{i^{\ell+1}, j^{\ell+1}, d^{\ell+1}} = \begin{cases} \left[\frac{\partial \mathcal{L}}{\partial f(X^{\ell+1})}\right]_{i^{\ell+1}, j^{\ell+1}, d^{\ell+1}} & \text{if } X^{\ell+1}_{i^{\ell+1}, j^{\ell+1}, d^{\ell+1}} \geq 0 \\ 0 \text{ otherwise} \end{cases} \qquad (19)$$

### E.1.3  POOLING

If we take into account the pooling operation, we must change some of the gradients. Indeed, a pixel that hasn't been selected during pooling has no impact on the final loss, thus should have a gradient equal to 0. We will focus on the case of Max Pooling (Average Pooling relies on similar idea). To state a formal relation, we will use the notations of Section D.2.2: an element in the output of the layer, the tensor $f(X^{\ell+1})$, is located by the triplet $(i^{\ell+1}, j^{\ell+1}, d^{\ell+1})$. The tensor after pooling is noted $\tilde{X}^{\ell+1}$ and its points are located by the triplet $(\tilde{i}^{\ell+1}, \tilde{j}^{\ell+1}, \tilde{d}^{\ell+1})$. During backpropagation, after the calculation of $\frac{\partial \mathcal{L}}{\partial \tilde{X}^{\ell+1}}$, some of the derivatives of $f(X^{\ell+1})$ should be set to zero with the following rule:

$$\left[\frac{\partial \mathcal{L}}{\partial f(X^{\ell+1})}\right]_{i^{\ell+1}, j^{\ell+1}, d^{\ell+1}} = \begin{cases} \left[\frac{\partial \mathcal{L}}{\partial \tilde{X}^{\ell+1}}\right]_{\tilde{i}^{\ell+1}, \tilde{j}^{\ell+1}, \tilde{d}^{\ell+1}} & \text{if } (i^{\ell+1}, j^{\ell+1}, d^{\ell+1}) \text{ was selected during pooling} \\ 0 \text{ otherwise} \end{cases}$$

$$(20)$$

### E.2 Quantum Algorithm for Backpropagation

In this section, we want to give a quantum algorithm to perform backrpopagation on a layer $\ell$, and detail the impact on the derivatives, given by the following diagram:

$$
\begin{cases} \frac{\partial \mathcal{L}}{\partial X^\ell} \\ \frac{\partial \mathcal{L}}{\partial F^\ell} \end{cases} \leftarrow \frac{\partial \mathcal{L}}{\partial \overline{X}^{\ell+1}} \leftarrow \frac{\partial \mathcal{L}}{\partial f(\overline{X}^{\ell+1})} \leftarrow \frac{\partial \mathcal{L}}{\partial \mathcal{X}^{\ell+1}} \leftarrow \frac{\partial \mathcal{L}}{\partial \tilde{\mathcal{X}}^{\ell+1}} = \frac{\partial \mathcal{L}}{\partial X^{\ell+1}} \tag{21}
$$

We assume that backpropagation has been done on layer $\ell+1$. This means in particular that $\frac{\partial \mathcal{L}}{\partial X^{\ell+1}}$ is stored in QRAM. However, as shown on Diagram (21), $\frac{\partial \mathcal{L}}{\partial X^{\ell+1}}$ corresponds formally to $\frac{\partial \mathcal{L}}{\partial \tilde{\mathcal{X}}^{\ell+1}}$, and not $\frac{\partial \mathcal{L}}{\partial \overline{X}^{\ell+1}}$. Therefore, we will have to modify the values stored in QRAM to take into account non linearity, tomography and pooling. We will first consider how to implement $\frac{\partial \mathcal{L}}{\partial X^\ell}$ and $\frac{\partial \mathcal{L}}{\partial F^\ell}$ through backpropagation, considering only convolution product, as if $\frac{\partial \mathcal{L}}{\partial \overline{X}^{\ell+1}}$ and $\frac{\partial \mathcal{L}}{\partial X^{\ell+1}}$ where the same. Then we will detail how to simply modify $\frac{\partial \mathcal{L}}{\partial X^{\ell+1}}$ *a priori*, by setting some of its values to 0.

#### E.2.1 Quantum Convolution Product

In this section we consider only the quantum convolution product without non linearity, tomography nor pooling, hence writing its output directly as $X^{\ell+1}$. Regarding derivatives, the quantum convolution product is equivalent to the classical one. Gradient relations (17) and (18) remain the same. Note that the $\epsilon$-approximation from Section D.1.2 doesn't participate in gradient considerations.

The gradient relations being the same, we still have to specify the quantum algorithm that implements the backpropagation and outputs classical description of $\frac{\partial \mathcal{L}}{\partial X^\ell}$ and $\frac{\partial \mathcal{L}}{\partial F^\ell}$. We have seen that the two main calculations (17) and (18) are in fact matrix-matrix multiplications both involving $\frac{\partial \mathcal{L}}{\partial Y^{\ell+1}}$, the reshaped form of $\frac{\partial \mathcal{L}}{\partial X^{\ell+1}}$. For each, the classical running time is $O(H^{\ell+1}W^{\ell+1}D^{\ell+1}HWD^\ell)$. We know from Theorem F.7 and Theorem G.1 a quantum algorithm to perform efficiently a matrix-vector multiplication and return a classical state with $\ell_\infty$ norm guarantees. For a matrix $V$ and a vector $b$, both accessible from the QRAM, the running time to perform this operation is $O\left(\frac{\mu(V)\kappa(V)\log 1/\delta}{\delta^2}\right)$, where $\kappa(V)$ is the condition number of the matrix and $\mu(V)$ is a matrix parameter defined in Equation (5). Precision parameter $\delta > 0$ is the error committed in the approximation for both Theorems F.7 and G.1.

We can therefore apply theses theorems to perform matrix-matrix multiplications, by simply decomposing them in several matrix-vector multiplications. For instance, in Equation (17), the matrix could be $(A^\ell)^T$ and the different vectors would be each column of $\frac{\partial L}{\partial Y^{\ell+1}}$. The global running time to perform quantumly Equation (17) is obtained by replacing $\mu(V)$ by $\mu(\frac{\partial \mathcal{L}}{\partial Y^{\ell+1}}) + \mu(A^\ell)$ and $\kappa(V)$ by $\kappa((A^\ell)^T \cdot \frac{\partial \mathcal{L}}{\partial Y^{\ell+1}})$. Likewise, for Equation (18), we have $\mu(\frac{\partial \mathcal{L}}{\partial Y^{\ell+1}}) + \mu(F^\ell)$ and $\kappa(\frac{\partial \mathcal{L}}{\partial Y^{\ell+1}} \cdot (F^\ell)^T)$.

Note that the dimension of the matrix doesn't appear in the running time since we tolerate a $\ell_\infty$ norm guarantee for the error, instead of a $\ell_2$ guarantee (see Section G for details). The reason why $\ell_\infty$ tomography is the right approximation here is because the result of these linear algebra operations are rows of the gradient matrices, that are not vectors in an euclidean space, but a series of numbers for which we want to be $\delta$-close to the exact values. See next section for more details.

It is a open question to see if one can apply the same sub-sampling technique as in the forward pass (Section D.1) and sample only the highest derivatives of $\frac{\partial \mathcal{L}}{\partial X^\ell}$, to reduce the computation cost while maintaining a good optimization. We then have to understand which elements of $\frac{\partial \mathcal{L}}{\partial X^{\ell+1}}$ must be set to zero to take into account the effects the non linearity, tomography and pooling.

#### E.2.2 Quantum Non Linearity and Tomography

To include the impact of the non linearity, one could apply the same rule as in (19), and simply replace ReLu by capReLu. After the non linearity, we obtain $f(\overline{X}^{\ell+1})$, and the gradient relation would be given by

$$\left[\frac{\partial \mathcal{L}}{\partial \overline{X}^{\ell+1}}\right]_{i^{\ell+1},j^{\ell+1},d^{\ell+1}} = \begin{cases} \left[\frac{\partial \mathcal{L}}{\partial f(\overline{X}^{\ell+1})}\right]_{i^{\ell+1},j^{\ell+1},d^{\ell+1}} & \text{if } 0 \leq \overline{X}^{\ell+1}_{i^{\ell+1},j^{\ell+1},d^{\ell+1}} \leq C \\ 0 \text{ otherwise} \end{cases} \tag{22}$$

If an element of $\overline{X}^{\ell+1}$ was negative or bigger than the cap $C$, its derivative should be zero during the backpropagation. However, this operation was performed in quantum superposition. In the quantum algorithm, one cannot record at which positions $(i^{\ell+1}, j^{\ell+1}, d^{\ell+1})$ the activation function was selective or not. The gradient relation (22) cannot be implemented *a posteriori*. We provide a partial solution to this problem, using the fact that quantum tomography must also be taken into account for some derivatives. Indeed, only the points $(i^{\ell+1}, j^{\ell+1}, d^{\ell+1})$ that have been sampled should have an impact on the gradient of the loss. Therefore we replace the previous relation by

$$\left[\frac{\partial \mathcal{L}}{\partial \overline{X}^{\ell+1}}\right]_{i^{\ell+1},j^{\ell+1},d^{\ell+1}} = \begin{cases} \left[\frac{\partial \mathcal{L}}{\partial \mathcal{X}^{\ell+1}}\right]_{i^{\ell+1},j^{\ell+1},d^{\ell+1}} & \text{if } (i^{\ell+1}, j^{\ell+1}, d^{\ell+1}) \text{ was sampled} \\ 0 \text{ otherwise} \end{cases} \tag{23}$$

Nonetheless, we can argue that this approximation will be tolerable. In the first case where $\overline{X}^{\ell+1}_{i^{\ell+1},j^{\ell+1},d^{\ell+1}} < 0$, the derivatives can not be set to zero as they should. But in practice, their values will be zero after the activation function and such points would not have a chance to be sampled. In conclusion their derivatives would be zero as required. In the other case, where $\overline{X}^{\ell+1}_{i^{\ell+1},j^{\ell+1},d^{\ell+1}} > C$, the derivatives can not be set to zero as well but the points have a high probability of being sampled. Therefore their derivative will remain unchanged, as if we were using a ReLu instead of a capReLu. However in cases where the cap $C$ is high enough, this shouldn't be a source of disadvantage in practice.

### E.2.3 QUANTUM POOLING

From relation (23), we can take into account the impact of quantum pooling (see Section D.2.2) on the derivatives. This case is easier since one can record the selected positions during the QRAM update. Therefore, applying the backpropagation is similar to the classical setting with Equation (20).

$$\left[\frac{\partial \mathcal{L}}{\partial \mathcal{X}^{\ell+1}}\right]_{i^{\ell+1},j^{\ell+1},d^{\ell+1}} = \begin{cases} \left[\frac{\partial \mathcal{L}}{\partial \overline{\mathcal{X}}^{\ell+1}}\right]_{\tilde{i}^{\ell+1},\tilde{j}^{\ell+1},\tilde{d}^{\ell+1}} & \text{if } (i^{\ell+1}, j^{\ell+1}, d^{\ell+1}) \text{ was selected during pooling} \\ 0 \text{ otherwise} \end{cases}$$
$$\tag{24}$$

Note that we know $\frac{\partial \mathcal{L}}{\partial \overline{\mathcal{X}}^{\ell+1}}$ as it is equal to $\frac{\partial \mathcal{L}}{\partial X^{\ell+1}}$, the gradient with respect to the input of layer $\ell+1$, known by assumption and stored in the QRAM.

### E.3 CONCLUSION AND RUNNING TIME

In conclusion, given $\frac{\partial \mathcal{L}}{\partial Y^{\ell+1}}$ in the QRAM, the quantum backpropagation first consists in applying the relations (24) followed by (23). The effective gradient now take into account non linearity, tomography and pooling that occurred during layer $\ell$. We can know use apply the quantum algorithm for matrix-matrix multiplication that implements relations (18) and (17).

Note that the steps in Algorithm 2 could also be reversed: during backpropagation of layer $\ell + 1$, when storing values for each elements of $\frac{\partial \mathcal{L}}{\partial Y^{\ell+1}}$ in the QRAM, one can already take into account (24) and (23) of layer $\ell$. In this case we directly store $\frac{\partial \mathcal{L}}{\partial \overline{X}^{\ell+1}}$, at no supplementary cost.

Therefore, the running time of the quantum backpropagation for one layer $\ell$, given as Algorithm 2, corresponds to the sum of the running times of the circuits for implementing relations (17) and (18). We finally obtain

$$O\left(\left(\left(\mu(A^\ell) + \mu(\tfrac{\partial\mathcal{L}}{\partial Y^{\ell+1}})\right)\kappa((A^\ell)^T \cdot \tfrac{\partial\mathcal{L}}{\partial Y^{\ell+1}}) + \left(\mu(\tfrac{\partial\mathcal{L}}{\partial Y^{\ell+1}}) + \mu(F^\ell)\right)\kappa(\tfrac{\partial\mathcal{L}}{\partial Y^{\ell+1}} \cdot (F^\ell)^T)\right)\tfrac{\log 1/\delta}{\delta^2}\right),$$

which can be rewritten as

$$O\left(\left(\left(\mu(A^\ell) + \mu(\frac{\partial\mathcal{L}}{\partial Y^{\ell+1}})\right)\kappa(\frac{\partial\mathcal{L}}{\partial F^\ell}) + \left(\mu(\frac{\partial\mathcal{L}}{\partial Y^{\ell+1}}) + \mu(F^\ell)\right)\kappa(\frac{\partial\mathcal{L}}{\partial Y^\ell})\right)\frac{\log 1/\delta}{\delta^2}\right) \qquad (25)$$

Besides storing $\frac{\partial\mathcal{L}}{\partial X^\ell}$, the main output is a classical description of $\frac{\partial\mathcal{L}}{\partial F^\ell}$, necessary to perform gradient descent of the parameters of $F^\ell$. In the Appendix (Section E.4), which details the impact of the quantum backpropagation compared to the classical case, which can be reduced to a simple noise addition during the gradient descent.

### E.4 Quantum Gradient Descent and Classical equivalence

In this part we will see the impact of the quantum backpropagation compared to the classical case, which can be reduced to a simple noise addition during the gradient descent. Recall that gradient descent, in our case, would consist in applying the following update rule $F^\ell \leftarrow F^\ell - \lambda\frac{\partial\mathcal{L}}{\partial F^\ell}$ with the learning rate $\lambda$.

Let's note $x = \frac{\partial\mathcal{L}}{\partial F^\ell}$ and its elements $x_{s,q} = \frac{\partial\mathcal{L}}{\partial F^\ell_{s,q}}$. From the first result of Theorem F.7 with error $\delta < 0$, and the tomography procedure from Theorem G.1, with same error $\delta$, we can obtain a classical description of $\frac{\overline{x}}{\|\overline{x}\|_2}$ with $\ell_\infty$ norm guarantee, such that:

$$\left\|\frac{\overline{x}}{\|\overline{x}\|_2} - \frac{x}{\|x\|_2}\right\|_\infty \leq \delta$$

in time $\widetilde{O}(\frac{\kappa(V)\mu(V)\log(\delta)}{\delta^2})$, where we note $V$ is the matrix stored in the QRAM that allows to obtain $x$, as explained in Section E.2. The $\ell_\infty$ norm tomography is used so that the error $\delta$ is at most the same for each component

$$\forall(s,q), \left|\frac{\overline{x_{s,q}}}{\|\overline{x}\|_2} - \frac{x_{s,q}}{\|x\|_2}\right| \leq \delta$$

From the second result of the Theorem F.7 we can also obtain an estimate $\|\overline{x}\|_2$ of the norm, for the same error $\delta$, such that

$$|\,\|\overline{x}\|_2 - \|x\|_2\,| \leq \delta\,\|x\|_2$$

in time $\widetilde{O}(\frac{\kappa(V)\mu(V)}{\delta}\log(\delta))$ (which does not affect the overall asymptotic running time). Using both results we can obtain an unnormalized state close to $x$ such that, by the triangular inequality

$$\|\overline{x} - x\|_\infty = \left\|\frac{\overline{x}}{\|\overline{x}\|_2}\|\overline{x}\|_2 - \frac{x}{\|x\|_2}\|x\|_2\right\|_\infty$$

$$\leq \left\|\frac{\overline{x}}{\|\overline{x}\|_2}\|\overline{x}\|_2 - \frac{\overline{x}}{\|\overline{x}\|_2}\|x\|_2\right\|_\infty + \left\|\frac{\overline{x}}{\|\overline{x}\|_2}\|x\|_2 - \frac{x}{\|x\|_2}\|x\|_2\right\|_\infty$$

$$\leq 1 \cdot |\,\|\overline{x}\|_2 - \|x\|_2\,| + \|x\|_2 \cdot \left\|\frac{\overline{x}}{\|\overline{x}\|_2} - \frac{x}{\|x\|_2}\right\|_\infty$$

$$\leq \delta\,\|x\|_2 + \|\overline{x}\|_2\,\delta \leq 2\delta\,\|x\|_2$$

in time $\widetilde{O}(\frac{\kappa(V)\mu(V)\log(\delta)}{\delta^2})$. In conclusion, with $\ell_\infty$ norm guarantee, having also access to the norm of the result is costless.

Finally, the noisy gradient descent update rule, expressed as $F^\ell_{s,q} \leftarrow F^\ell_{s,q} - \lambda\overline{\frac{\partial\mathcal{L}}{\partial F^\ell_{s,q}}}$ can written in the worst case with

$$\overline{\frac{\partial\mathcal{L}}{\partial F^\ell_{s,q}}} = \frac{\partial\mathcal{L}}{\partial F^\ell_{s,q}} \pm 2\delta\left\|\frac{\partial\mathcal{L}}{\partial F^\ell}\right\|_2 \qquad (26)$$

To summarize, using the quantum linear algebra from Theroem F.7 with $\ell_\infty$ norm tomography from Theroem G.1, both with error $\delta$, along with norm estimation with relative error $\delta$ too, we can obtain classically the unnormalized values $\overline{\frac{\partial\mathcal{L}}{\partial F^\ell}}$ such that $\left\|\overline{\frac{\partial\mathcal{L}}{\partial F^\ell}} - \frac{\partial\mathcal{L}}{\partial F^\ell}\right\|_\infty \leq 2\delta\left\|\frac{\partial\mathcal{L}}{\partial F^\ell}\right\|_2$ or equivalently

$$\forall(s,q), \left|\overline{\frac{\partial\mathcal{L}}{\partial F^\ell_{s,q}}} - \frac{\partial\mathcal{L}}{\partial F^\ell_{s,q}}\right| \leq 2\delta\left\|\frac{\partial\mathcal{L}}{\partial F^\ell}\right\|_2 \qquad (27)$$

Therefore the gradient descent update rule in the quantum case becomes $F_{s,q}^{\ell} \leftarrow F_{s,q}^{\ell} - \lambda \overline{\frac{\partial \mathcal{L}}{\partial F_{s,q}^{\ell}}}$, which in the worst case becomes

$$F_{s,q}^{\ell} \leftarrow F_{s,q}^{\ell} - \lambda \left( \frac{\partial \mathcal{L}}{\partial F_{s,q}^{\ell}} \pm 2\delta \left\| \frac{\partial \mathcal{L}}{\partial F^{\ell}} \right\|_2 \right) \tag{28}$$

This proves the Theorem E.1. This update rule can be simulated by the addition of a random relative noise given as a gaussian centered on 0, with standard deviation equal to $\delta$. This is how we will simulate quantum backpropagation in the Numerical Simulations.

Compared to the classical update rule, this corresponds to the addition of noise during the optimization step. This noise decreases as $\left\| \frac{\partial \mathcal{L}}{\partial F^{\ell}} \right\|_2$, which is expected to happen while converging. Recall that the gradient descent is already a stochastic process. Therefore, we expect that such noise, with acceptable values of $\delta$, will not disturb the convergence of the gradient, as the following numerical simulations tend to confirm.

## APPENDIX F    PRELIMINARIES IN QUANTUM INFORMATION

We introduce a basic and broad-audience quantum information background necessary for this work. For a more detailed introduction we recommend Nielsen & Chuang (2002a).

### F.1    QUANTUM INFORMATION

**Quantum Bits and Quantum Registers:**    The bit is the most basic unit of classical information. It can be either in state 0 or 1. Similarly a quantum bit or *qubit*, is a quantum system that can be is state $|0\rangle$, $|1\rangle$ (the *braket* notation $|\cdot\rangle$ is a reminder that the bit considered is a quantum system) or in superposition of both states $\alpha |0\rangle + \beta |1\rangle$ with coefficients $\alpha, \beta \in \mathbb{C}$ such that $|\alpha|^2 + |\beta|^2 = 1$. The *amplitudes* $\alpha$ and $\beta$ are linked to the probabilities of observing either 0 or 1 when *measuring* the qubit, since $P(0) = |\alpha|^2$ and $P(1) = |\beta|^2$.

Before the measurement, any superposition is possible, which gives quantum information special abilities in terms of computation. With $n$ qubits, the $2^n$ possible binary combinations can exist simultaneously, each with a specific amplitude. For instance we can consider an uniform distribution $\frac{1}{\sqrt{n}} \sum_{i=0}^{2^n-1} |i\rangle$ where $|i\rangle$ represents the $i^{th}$ binary combination (e.g. $|01\cdots1001\rangle$). Multiple qubits together are often called a *quantum register*.

In its most general formulation, a quantum state with $n$ qubits can be seen as vector in a complex Hilbert space of dimension $2^n$. This vector must be normalized under $\ell_2$-norm, to guarantee that the squared amplitudes sum to 1.

**Quantum Computation:**    To process qubits and therefore quantum registers, we use quantum gates. These gates are *unitary operators* in the Hilbert space as they should map unit-norm vectors to unit-norm vectors. Formally, we can see a quantum gate acting on $n$ qubits as a matrix $U \in \mathbb{C}^{2^n}$ such that $UU^{\dagger} = U^{\dagger}U = I$, where $U^{\dagger}$ is the conjugate transpose of $U$. Some basic single qubit gates includes the NOT gate $\begin{pmatrix} 0 & 1 \\ 1 & 0 \end{pmatrix}$ that inverts $|0\rangle$ and $|1\rangle$, or the Hadamard gate $\frac{1}{\sqrt{2}} \begin{pmatrix} 1 & 1 \\ 1 & -1 \end{pmatrix}$ that maps $|0\rangle \mapsto \frac{1}{\sqrt{2}}(|0\rangle + |1\rangle)$ and $|1\rangle \mapsto \frac{1}{\sqrt{2}}(|0\rangle - |1\rangle)$, creating the quantum superposition.

Finally, multiple qubits gates exist, such as the Controlled-NOT that applies a NOT gate on a target qubit conditioned on the state of a control qubit.

The main advantage of quantum gates is their ability to be applied to a superposition of inputs. Indeed, given a gate $U$ such that $U |x\rangle \mapsto |f(x)\rangle$, we can apply it to all possible combinations of $x$ at once $U(\frac{1}{C} \sum_x |x\rangle) \mapsto \frac{1}{C} \sum_x |f(x)\rangle$.

We now state some primitive quantum circuits, which we will use in our algorithm:

For two integers $i$ and $j$, we can check their equality with the mapping $|i\rangle |j\rangle |0\rangle \mapsto |i\rangle |j\rangle |[i = j]\rangle$. For two real value numbers $a > 0$ and $\delta > 0$, we can compare them using $|a\rangle |\delta\rangle |0\rangle \mapsto |a\rangle |\delta\rangle |[a \leq \delta]\rangle$. Finally, for a real value numbers $a > 0$, we can obtain its square $|a\rangle |0\rangle \mapsto |a\rangle |a^2\rangle$.

Note that these circuits are basically a reversible version of the classical ones and are linear in the number of qubits used to encode the input values.

## F.2 QUANTUM SUBROUTINES FOR DATA ENCODING

Knowing some basic principles of quantum information, the next step is to understand how data can be efficiently encoded using quantum states. While several approaches could exist, we present the most common one called *amplitude encoding*, which leads to interesting and efficient applications.

Let $x \in \mathbb{R}^d$ be a vector with components $(x_1, \cdots, x_d)$. Using only $\lceil \log(d) \rceil$ qubits, we can form $|x\rangle$, the quantum state encoding $x$, given by $|x\rangle = \frac{1}{\|x\|} \sum_{j=0}^{d-1} x_j |j\rangle$. We see that the $j^{th}$ component $x_j$ becomes the amplitude of $|j\rangle$, the $j^{th}$ binary combination (or equivalently the $j^{th}$ vector in the standard basis). Each amplitude must be divided by $\|x\|$ to preserve the unit $\ell_2$-norm of $|x\rangle$.

Similarly, for a matrix $A \in \mathbb{R}^{n \times d}$ or equivalently for $n$ vectors $A_i$ for $i \in [n]$, we can express each row of $A$ as $|A_i\rangle = \frac{1}{\|A_i\|} \sum_{i=0}^{d-1} A_{ij} |j\rangle$.

We can now explain an important definition, the ability to have *quantum access* to a matrix. This will be a requirements for many algorithms.

**Definition 4** *[Quantum Access to Data]*
 *We say that we have quantum access to a matrix $A \in \mathbb{R}^{n \times d}$ if there exist a procedure to perform the following mapping, for $i \in [n]$, in time $T$:*

- $|i\rangle |0\rangle \mapsto |i\rangle |A_i\rangle$

- $|0\rangle \mapsto \frac{1}{\|A\|_F} \sum_i \|A_i\| |i\rangle$

By using appropriate data structures the first mapping can be reduced to the ability to perform a mapping of the form $|i\rangle |j\rangle |0\rangle \mapsto |i\rangle |j\rangle |A_{ij}\rangle$. The second requirement can be replaced by the ability of performing $|i\rangle |0\rangle \mapsto |i\rangle |\|A_i\|\rangle$ or to just have the knowledge of each norm. Therefore, using matrices such that all rows $A_i$ have the same norm makes it simpler to obtain the quantum access.

The time or complexity $T$ necessary for the quantum access can be reduced to polylogarithmic dependence in $n$ and $d$ if we consider the access to a Quantum Memory or *QRAM*. The QRAM Kerenidis & Prakash (2017a) is a specific data structure from which a quantum circuit can allow quantum access to data in time $O(\log(nd))$.

**Theorem F.1 (QRAM data structure, see Kerenidis & Prakash (2017a))** *Let $A \in \mathbb{R}^{n \times d}$, there is a data structure to store the rows of $A$ such that,*

1. *The time to insert, update or delete a single entry $A_{ij}$ is $O(\log^2(n))$.*

2. *A quantum algorithm with access to the data structure can perform the following unitaries in time $T = O(\log^2 n)$.*

    (a) *$|i\rangle |0\rangle \rightarrow |i\rangle |A_i\rangle$ for $i \in [n]$.*
    (b) *$|0\rangle \rightarrow \sum_{i \in [n]} \|A_i\| |i\rangle$.*

We now state important methods for processing the quantum information. Their goal is to store some information alternatively in the quantum state's amplitude or in the quantum register as a bitstring.

**Theorem F.2** *[Amplitude Amplification and Estimation Brassard et al. (2002)] Given a unitary operator $U$ such that $U : |0\rangle \mapsto \sqrt{p} |y\rangle |0\rangle + \sqrt{1-p} |y^\perp\rangle |1\rangle$ in time $T$, where $p > 0$ is the probability of measuring "0", it is possible to obtain the state $|y\rangle |0\rangle$ using $O(\frac{T}{\sqrt{p}})$ queries to $U$, or to estimate $p$ with relative error $\delta$ using $O(\frac{T}{\delta \sqrt{p}})$ queries to $U$.*

**Theorem F.3** *[Conditional Rotation] Given the quantum state $|a\rangle$, with $a \in [-1, 1]$, it is possible to perform $|a\rangle |0\rangle \mapsto |a\rangle (a |0\rangle + \sqrt{1-a} |1\rangle)$ with complexity $\widetilde{O}(1)$.*

Using Theorem F.3 followed by Theorem F.2, it then possible to transform the state $\frac{1}{\sqrt{d}} \sum_{j=0}^{d-1} |x_j\rangle$ into $\frac{1}{\|x\|} \sum_{j=0}^{d-1} x_j |x_j\rangle$.

In addition to amplitude estimation, we will make use of a tool developed in Wiebe et al. (2014a) to boost the probability of getting a good estimate for the inner product required for the quantum convolution algorithm. In high level, we take multiple copies of the estimator from the amplitude estimation procedure, compute the median, and reverse the circuit to get rid of the garbage. Here we provide a theorem with respect to time and not query complexity.

**Theorem F.4 (Median Evaluation, see Wiebe et al. (2014a))** *Let $\mathcal{U}$ be a unitary operation that maps*

$$\mathcal{U} : |0^{\otimes n}\rangle \mapsto \sqrt{a} |x, 1\rangle + \sqrt{1-a} |G, 0\rangle$$

*for some $1/2 < a \leq 1$ in time $T$. Then there exists a quantum algorithm that, for any $\Delta > 0$ and for any $1/2 < a_0 \leq a$, produces a state $|\Psi\rangle$ such that $\| |\Psi\rangle - |0^{\otimes nL}\rangle |x\rangle \| \leq \sqrt{2\Delta}$ for some integer $L$, in time*

$$2T \left\lceil \frac{\ln(1/\Delta)}{2 \left(|a_0| - \frac{1}{2}\right)^2} \right\rceil.$$

### F.3 QUANTUM SUBROUTINES FOR LINEAR ALGEBRA

In the recent years, as the field of quantum machine learning grew, its "toolkit" for linear algebra algorithms has become important enough to allow the development of many quantum machine learning algorithms. We introduce here the important subroutines for this work, without detailing the circuits or the algorithms.

**Definition 5** *For a matrix $A$, the parameter $\mu(A)$ is defined by $\mu(A) = \min_{p \in [0,1]} \left( \|A\|_F, \sqrt{s_{2p}(A) s_{2(1-p)}(A^T)} \right)$ where $s_p(A) = \max_i(\|A_i\|_p^p)$.*

The next theorems allow to compute the distance between vectors encoded as quantum states, and use this idea to perform the $k$-means algorithm.

**Theorem F.5** *[Quantum Distance Estimation Wiebe et al. (2014b); Kerenidis et al. (2019)] Given quantum access in time $T$ to two matrices $U$ and $V$ with rows $u_i$ and $v_j$ of dimension $d$, there is a quantum algorithm that, for any pair $(i,j)$, performs the following mapping $|i\rangle |j\rangle |0\rangle \mapsto |i\rangle |j\rangle |\overline{d^2(u_i, v_j)}\rangle$, estimating the euclidean distance between $u_i$ and $v_j$ with precision $|\overline{d^2(u_i, v_j)} - d^2(u_i, v_j)| \leq \epsilon$ for any $\epsilon > 0$. The algorithm has a running time given by $\widetilde{O}(T\eta/\epsilon)$, where $\eta = \max_{ij}(\|u_i\| \|v_j\|)$, assuming that $\min_i(\|u_i\|) = \min_i(\|v_i\|) = 1$.*

**Theorem F.6** *[Quantum k-means clustering Kerenidis et al. (2019)]*
*Given quantum access in time $T$ to a dataset $V \in \mathbb{R}^{n \times d}$, there is a quantum algorithm that outputs with high probability $k$ centroids $c_1, \cdots, c_k$ that are consistent with the output of the $k$-means algorithm with noise $\delta > 0$, in time $\widetilde{O}(T \times (kd\frac{\eta(V)}{\delta^2}\kappa(V)(\mu(V) + k\frac{\eta(V)}{\delta}) + k^2 \frac{\eta(V)^{1.5}}{\delta^2}\kappa(V)\mu(V)))$ per iteration.*

**Definition 6** *For a matrix $V \in \mathbb{R}^{n \times d}$, its parameter $\eta(V)$ is defined as as $\frac{\max_i(\|v_i\|^2)}{\min_i(\|v_i\|^2)}$, or as $\max_i(\|v_i\|^2)$ assuming $\min_i(\|v_i\|) = 1$.*

In theorem F.6, the other parameters in the running time can be interpreted as follows : $\delta$ is the precision in the estimation of the distances, but also in the estimation of the position of the centroids. $\kappa(V)$ is the condition number of $V$ and $\mu(V)$ is defined above (Definition 5). Finally, in the case of *well clusterable datasets*, which should be the case when we will apply $k$-means during spectral clustering, the running simplifies to $\widetilde{O}(T \times (k^2 d\frac{\eta(V)^{2.5}}{\delta^3} + k^{2.5}\frac{\eta(V)^2}{\delta^3}))$.

Note that the dependence in $n$ is hidden in the time $T$ to load the data. This dependence becomes polylogarithmic in $n$ if we assume access to a QRAM.

**Theorem F.7 (Quantum Matrix Operations, Chakraborty et al. (2018) )** *Let $M \in \mathbb{R}^{d \times d}$ and $x \in \mathbb{R}^d$. Let $\delta_1, \delta_2 > 0$. If $M$ is stored in appropriate QRAM data structures and the time to prepare $|x\rangle$ is $T_x$, then there exist quantum algorithms that with probability at least $1 - 1/poly(d)$ return*

1. *A state $|z\rangle$ such that $\||z\rangle - |Mx\rangle\|_2 \leq \delta_1$ in time $\widetilde{O}((\kappa(M)\mu(M) + T_x\kappa(M))\log(1/\delta_1))$. Note that this also implies $\||z\rangle - |\widetilde{M}x\rangle\|_\infty \leq \delta_1$*

2. *Norm estimate $z \in (1 \pm \delta_2)\|Mx\|_2$, with relative error $\delta_2$, in time $\widetilde{O}(T_x \frac{\kappa(M)\mu(M)}{\delta_2}\log(1/\delta_1))$.*

The linear algebra procedures above can also be applied to any rectangular matrix $V \in \mathbb{R}^{n \times d}$ by considering instead the symmetric matrix $\overline{V} = \begin{pmatrix} 0 & V \\ V^T & 0 \end{pmatrix}$.

## APPENDIX G  ALGORITHM AND PROOF FOR $\ell_\infty$ NORM TOMOGRAPHY

Finally, we present a logarithmic time algorithm for vector state tomography that will be used to re-cover classical information from the quantum states with $\ell_\infty$ norm guarantee. Given a unitary $U$ that produces a quantum state $|x\rangle = \frac{1}{\|x\|_2} \sum_{j=0}^{d-1} x_j |j\rangle$, by calling $O(\log d/\delta^2)$ times $U$, the tomography algorithm is able to reconstruct a vector $\widetilde{X}$ that approximates $|x\rangle$ with $\ell_\infty$ norm guarantee, such that $\left\||\widetilde{X}\rangle - |x\rangle\right\|_\infty \leq \delta$, or equivalently that $\forall i \in [d], |x_i - \widetilde{X}_i| \leq \delta$. Such a tomography is of interest when the components $x_i$ of a quantum state are not the coordinates of an meaningful vector in some linear space, but just a series of values, such that we don't want an overall guarantee on the vector (which is the case with usual $\ell_2$ tomography) but a similar error guarantee for each component in the estimation.

**Theorem G.1 ($\ell_\infty$ Vector state tomography)** *Given access to unitary $U$ such that $U|0\rangle = |x\rangle$ and its controlled version in time $T(U)$, there is a tomography algorithm with time complexity $O(T(U)\frac{\log d}{\delta^2})$ that produces unit vector $\widetilde{X} \in \mathbb{R}^d$ such that $\left\|\widetilde{X} - x\right\|_\infty \leq \delta$ with probability at least $(1 - 1/poly(d))$.*

The proof of this theorem is similar to the proof of the $\ell_2$-norm tomography by Kerenidis & Prakash (2018). However the $\ell_\infty$ norm tomography introduced in this paper depends only logarithmically and not linearly in the dimension $d$. Note that in our case, $T(U)$ will be logarithmic in the dimension.

**Theorem G.2** *[$\ell_2$ Vector state tomography Kerenidis & Prakash (2018)] Given access to unitary $U$ such that $U|0\rangle = |x\rangle$ and its controlled version in time $T(U)$, there is an algorithm that allows to output a classical vector $\widetilde{X} \in \mathbb{R}^d$ with $\ell_2$-norm guarantee $\left\|\widetilde{X} - x\right\|_2 \leq \delta$ for any $\delta > 0$, in time $O(T(U) \times \frac{d\log(d)}{\delta^2})$.*

In the following we consider a quantum state $|x\rangle = \sum_{i \in [d]} x_i |i\rangle$, with $x \in \mathbb{R}^d$ and $\|x\|_2 = 1$.

The following version of the Chernoff Bound will be used for analysis of algorithm 3.

**Theorem G.3** *(Chernoff Bound) Let $X_j$, for $j \in [N]$, be independent random variables such that $X_j \in [0, 1]$ and let $X = \sum_{j \in [N]} X_j$. We have the three following inqualities:*

1. *For $0 < \beta < 1, \mathbb{P}[X < (1 - \beta)\mathbb{E}[X]] \leq e^{-\beta^2\mathbb{E}[X]/2}$*

2. *For $\beta > 0, \mathbb{P}[X > (1 + \beta)\mathbb{E}[X]] \leq e^{-\frac{\beta^2}{2+\beta}\mathbb{E}[X]}$*

3. *For $0 < \beta < 1, \mathbb{P}[|X - \mathbb{E}[X]| \geq \beta\mathbb{E}[X]] \leq e^{-\beta^2\mathbb{E}[X]/3}$, by composing 1. and 2.*

---

**Algorithm 3** $\ell_\infty$ norm tomography

---

**Require:** Error $\delta > 0$, access to unitary $U : |0\rangle \mapsto |x\rangle = \sum_{i \in [d]} x_i |i\rangle$, the controlled version of $U$, QRAM access.

**Ensure:** Classical vector $\widetilde{X} \in \mathbb{R}^d$, such that $\left\| \widetilde{X} \right\| = 1$ and $\left\| \widetilde{X} - x \right\|_\infty < \delta$.

1: Measure $N = \frac{36 \ln d}{\delta^2}$ copies of $|x\rangle$ in the standard basis and count $n_i$, the number of times the outcome $i$ is observed. Store $\sqrt{p_i} = \sqrt{n_i/N}$ in QRAM data structure.
2: Create $N = \frac{36 \ln d}{\delta^2}$ copies of the state $\frac{1}{\sqrt{2}} |0\rangle \sum_{i \in [d]} x_i |i\rangle + \frac{1}{\sqrt{2}} |1\rangle \sum_{i \in [d]} \sqrt{p_i} |i\rangle$.
3: Apply an Hadamard gate on the first qubit to obtain

$$|\phi\rangle = \frac{1}{2} \sum_{i \in [d]} \left( (x_i + \sqrt{p_i}) |0, i\rangle + (x_i - \sqrt{p_i}) |1, i\rangle \right)$$

4: Measure both registers of each copy in the standard basis, and count $n(0, i)$ the number of time the outcome $(0, i)$ is observed.
5: Set $\sigma(i) = +1$ if $n(0, i) > 0.4 N p_i$ and $\sigma(i) = -1$ otherwise.
6: Output the unit vector $\widetilde{X}$ such that $\forall i \in [N], \widetilde{X}_i = \sigma_i \sqrt{p_i}$

---

**Theorem G.4** *Algorithm 3 produces an estimate $\widetilde{X} \in \mathbb{R}^d$ such that $\left\| \widetilde{X} - x \right\|_\infty < (1 + \sqrt{2})\delta$ with probability at least $1 - \frac{1}{d^{0.83}}$.*

Proving $\left\| x - \widetilde{X} \right\|_\infty \leq O(\delta)$ is equivalent to show that for all $i \in [d]$, we have $|x_i - \widetilde{X}_i| = |x_i - \sigma(i)\sqrt{p_i}| \leq O(\delta)$. Let $S$ be the set of indices defined by $S = \{i \in [d]; |x_i| > \delta\}$. We will separate the proof for the two cases where $i \in S$ and $i \notin S$.

**Case 1 :** $i \in S$.

We will show that if $i \in S$, we correctly have $\sigma(i) = sgn(x_i)$ with high probability. Therefore we will need to bound $|x_i - \sigma(i)\sqrt{p_i}| = ||x_i| - \sqrt{p_i}|$.

We suppose that $x_i > 0$. The value of $\sigma(i)$ correctly determines $sgn(x_i)$ if the number of times we have measured $(0, i)$ at Step 4. is more than half of the outcomes, *i.e.* $n(0, i) > \frac{1}{2}\mathbb{E}[n(0, i)]$. If $x_i < 0$, the same arguments holds for $n(1, i)$. We consider the random variable that represents the outcome of a measurement on state $|\phi\rangle$. The Chernoff Bound, part 1 with $\beta = 1/2$ gives

$$\mathbb{P}[n(0, i) \leq \frac{1}{2}\mathbb{E}[n(0, i)]] \leq e^{-\mathbb{E}[n(0,i)]/8} \tag{29}$$

From the definition of $|\phi\rangle$ we have $\mathbb{E}[n(0, i)] = \frac{N}{4}(x_i + \sqrt{p_i})^2$. We will lower bound this value with the following argument.

For the $k^{th}$ measurement of $|x\rangle$, with $k \in [N]$, let $X_k$ be a random variable such that $X_k = 1$ if the outcome is $i$, and 0 otherwise. We define $X = \sum_{k \in [N]} X_k$. Note that $X = n_i = N p_i$ and $\mathbb{E}[X] = N x_i^2$. We can apply the Chernoff Bound, part 3 on $X$ for $\beta = 1/2$ to obtain,

$$\mathbb{P}[|X - \mathbb{E}[X]| \geq \mathbb{E}[X]/2] \leq e^{-\mathbb{E}[X]/12} \tag{30}$$

$$\mathbb{P}[|x_i^2 - p_i| \geq x_i^2/2] \leq e^{-N x_i^2/12}$$

We have $N = \frac{36 \ln d}{\delta^2}$ and by assumption $x_i^2 > \delta^2$ (since $i \in S$). Therefore,

$$\mathbb{P}[|x_i^2 - p_i| \geq x_i^2/2] \leq e^{-36 \ln d/12} = 1/d^3$$

This proves that the event $|x_i^2 - p_i| \leq x_i^2/2$ occurs with probability at least $1 - \frac{1}{d^3}$ if $i \in S$. This previous inequality is equivalent to $\sqrt{2p_i/3} \leq |x_i| \leq \sqrt{2p_i}$. Thus, with high probability we have $\mathbb{E}[n(0,i)] = \frac{N}{4}(x_i + \sqrt{p_i})^2 \geq 0.82Np_i$, since $\sqrt{2p_i/3} \leq |x_i|$. Moreover, since $|p_i| \leq x_i^2/2$, $\mathbb{E}[n(0,i)] \geq 0.82Nx_i^2/2 \geq 14.7 \ln d$. Therefore, equation equation 29 becomes

$$\mathbb{P}[n(0,i) \leq 0.41Np_i] \leq e^{-1.83 \ln d} = 1/d^{1.83}$$

We conclude that for $i \in S$, if $n(0,i) > 0.41Np_i$, the sign of $x_i$ is determined correctly by $\sigma(i)$ with high probability $1 - \frac{1}{d^{1.83}}$, as indicated in Step 5.

We finally show $|x_i - \sigma(i)\sqrt{p_i}| = ||x_i| - \sqrt{p_i}|$ is bounded. Again by the Chernoff Bound (3.) we have, for $0 < \beta < 1$:

$$\mathbb{P}[|x_i^2 - p_i| \geq \beta x_i^2] \leq e^{\beta^2 Nx_i^2/3}$$

By the identity $|x_i^2 - p_i| = (|x_i| - \sqrt{p_i})(|x_i| + \sqrt{p_i})$ we have

$$\mathbb{P}\left[\left||x_i| - \sqrt{p_i}\right| \geq \beta \frac{x_i^2}{|x_i| + \sqrt{p_i}}\right] \leq e^{\beta^2 Nx_i^2/3}$$

Since $\sqrt{p_i} > 0$, we have $\beta \frac{x_i^2}{|x_i|+\sqrt{p_i}} \leq \beta \frac{x_i^2}{|x_i|} = \beta|x_i|$, therefore $\mathbb{P}\left[\left||x_i| - \sqrt{p_i}\right| \geq \beta|x_i|\right] \leq e^{\beta^2 Nx_i^2/3}$. Finally, by chosing $\beta = \delta/|x_i| < 1$ we have

$$\mathbb{P}\left[\left||x_i| - \sqrt{p_i}\right| \geq \delta\right] \leq e^{36 \ln d/3} = 1/d^{12}$$

We conclude that, if $i \in S$, we have $|x_i - \tilde{X}_i| \leq \delta$ with high probability.

Since $|S| \leq d$, the probability for this result to be true for all $i \in S$ is $1 - \frac{1}{d^{0.83}}$. This can be proved by using the Union Bound on the correctness of $\sigma(i)$.

**Case 2 :** $i \notin S$.
   If $i \notin S$, we need to separate again in two cases. When the estimated sign is wrong, *i.e.* $\sigma(i) = -sgn(x_i)$, we have to bound $|x_i - \sigma(i)\sqrt{p_i}| = ||x_i| + \sqrt{p_i}|$. On the contrary, if it is correct, *i.e.* $\sigma(i) = sgn(x_i)$, we have to bound $|x_i - \sigma(i)\sqrt{p_i}| = ||x_i| - \sqrt{p_i}| \leq ||x_i| + \sqrt{p_i}|$. Therefore only one bound is necessary.

We use Chernoff Bound (2.) on the random variable X with $\beta > 0$ to obtain

$$\mathbb{P}[p_i > (1 + \beta)x_i^2] \leq e^{\frac{\beta^2}{2+\beta}Nx_i^2}$$

We chose $\beta = \delta^2/x_i^2$ and obtain $\mathbb{P}[p_i > x_i^2 + \delta^2] \leq e^{\frac{\delta^4}{3\delta^2}N} = 1/d^{12}$. Therefore, if $i \notin S$, with very high probability $1 - \frac{1}{d^{12}}$ we have $p_i \leq x_i^2 + \delta^2 \leq 2\delta^2$. We can conclude and bound the error:

$$|x_i - \tilde{X}_i| \leq ||x_i| + \sqrt{p_i}| \leq \delta + \sqrt{2}\delta = (1 + \sqrt{2})\delta$$

Since $|\overline{S}| \leq d$, the probability for this result to be true for all $i \notin S$ is $1 - \frac{1}{d^{11}}$. This follows from applying the Union Bound on the event $p_i > x_i^2 + \delta^2$.

## APPENDIX H    ADDITIONAL NUMERICAL SIMULATIONS

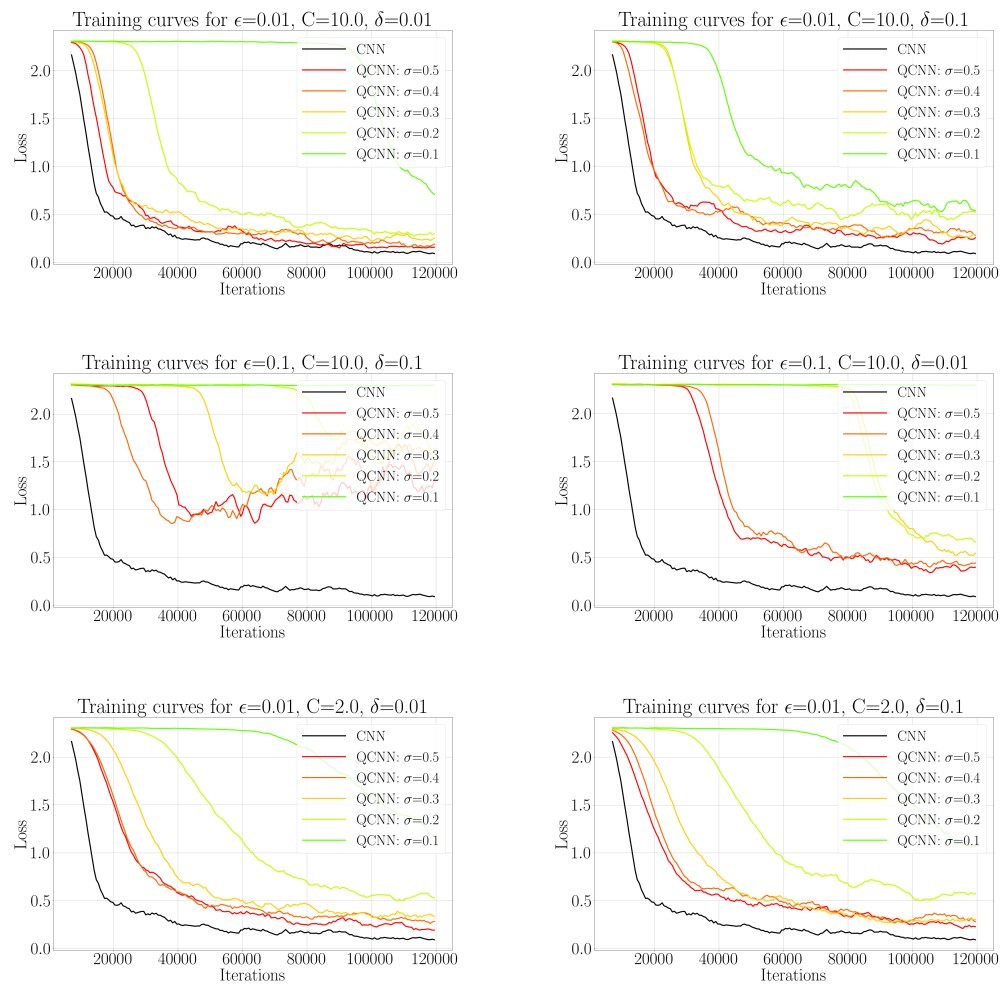

Figure 9: Numerical simulations of the training of the QCNN. These training curves represent the evolution of the Loss $\mathcal{L}$ as we iterate through the MNIST dataset. For each graph, the amplitude estimation error $\epsilon$ $(0.1, 0.01)$, the non linearity cap $C$ $(2, 10)$, and the backpropagation error $\delta$ $(0.1, 0.01)$ are fixed whereas the quantum sampling ratio $\sigma$ varies from 0.1 to 0.5. We can compare each training curve to the classical learning (CNN). Note that these training curves are smoothed, over windows of 12 steps, for readability.

In the following we report the classification results of the QCNN when applied on the test set (10.000 images). We distinguish to use cases: in Table 4 the QCNN has been trained quantumly as described in this paper, whereas in Table 5 we first have trained the classical CNN, then transferred the weights to the QCNN only for the classification. This second use case has a global running time worst than the first one, but we see it as another concrete application: quantum machine learning could be used only for faster classification from a classically generated model, which could be the case for high rate classification task (e.g. for autonomous systems, classification over many simultaneous inputs). We report the test loss and accuracy for different values of the sampling ratio $\sigma$, the amplitude estimation error $\epsilon$, and for the backpropagation noise $\delta$ in the first case. The cap $C$ is fixed at 10. These values must be compared to the classical CNN classification metrics, for which the loss is 0.129 and the accuracy is 96.1%. Note that we used a relatively small CNN and hence

the accuracy is just over $96\%$, lower than the best possible accuracy with larger CNN.

| QCNN Test - Classification | | | | | |
|---|---|---|---|---|---|
| $\sigma$ | $\epsilon$ | 0.01 | | 0.1 | |
| | $\delta$ | 0.01 | 0.1 | 0.01 | 0.1 |
| 0.1 | Loss | 0.519 | 0.773 | 2.30 | 2.30 |
| | Accuracy | 82.8% | 74.8% | 11.5% | 11.7% |
| 0.2 | Loss | 0.334 | 0.348 | 0.439 | 1.367 |
| | Accuracy | 89.5% | 89.0% | 86.2% | 54.1% |
| 0.3 | Loss | 0.213 | 0.314 | 0.381 | 0.762 |
| | Accuracy | 93.4% | 90.3% | 87.9% | 76.8% |
| 0.4 | Loss | 0.177 | 0.215 | 0.263 | 1.798 |
| | Accuracy | 94.7% | 93.3% | 91.8% | 34.9% |
| 0.5 | Loss | 0.142 | 0.211 | 0.337 | 1.457 |
| | Accuracy | 95.4% | 93.5% | 89.2% | 52.8% |

Table 4: QCNN trained with quantum backpropagation on MNIST dataset. With $C = 10$ fixed.

| QCNN Test - Classification | | | |
|---|---|---|---|
| $\sigma$ | $\epsilon$ | 0.01 | 0.1 |
| 0.1 | Loss | 1.07 | 1.33 |
| | Accuracy | 86.1% | 78.6% |
| 0.2 | Loss | 0.552 | 0.840 |
| | Accuracy | 92.8% | 86.5% |
| 0.3 | Loss | 0.391 | 0.706 |
| | Accuracy | 94,3% | 85.8% |
| 0.4 | Loss | 0.327 | 0.670 |
| | Accuracy | 94.4% | 84.0% |
| 0.5 | Loss | 0.163 | 0.292 |
| | Accuracy | 95.9% | 93.5% |

Table 5: QCNN created from a classical CNN trained on MNIST dataset. With $\delta = 0.01$ and $C = 10$ fixed.

