# OpenReview forum: "Quantum Algorithms for Deep Convolutional Neural Networks"
_ICLR.cc/2020/Conference — Accept (Poster)_

### Official Review · AnonReviewer3 · 2019-10-22
**Official Blind Review #3**

**Rating:** 6

**Review:**

This paper presents a quantum version of the convolutional neural networks. They derive equivalent versions the computation of both the convolution operation and the back-propagation in the quantum computing framework. The authors claim a potential exponential speed up in the computation in the size of the kernel which would make possible to process much bigger inputs or just speed up current tasks involving images mainly. They exemplify the method on MNIST using quantum artifacts simulation using PyTorch and show their method is competitive with SoA.

I think the paper is well written and provides a nice discussion about how quantum computing can be applied to CNNs. I appreciate that both the forward and backward passes are studied although most of the technical details are in appendices. So, I felt the paper  itself was a bit optimistic about the impact of quantum computing on CNNs.

Especially, I found the experimental section was missing details. I am not a quantum computing expert but I was a bit surprised that, in the context of CNNs, the introduction of quantum noise was just considered as introducing noise in the image and the gradient and then applying normal CNNs to the resulting image. Standard CNNs can probably deal with such noise and noisy gradient descent is not a big issue as such (it can even avoid local minima). But CNNs are notoriously known to reduce the number of weights in a network because of weight sharing. So, it is not all about making one convolution faster but also to compute invariant representations by sharing weights over different parts of the inputs. I would have been interested by a discussion about how quantum noise may impact this property and I didn't find this in the paper nor the appendices.

Also the author confess that the learning is not stable and results on MNIST to be the best they could get. I think it would be worth testing on large scale problems and see whether larger kernels with such noisy conditions would really improve the performance.

**Experience Assessment:**

I do not know much about this area.

**Review Assessment: Checking Correctness Of Derivations And Theory:**

I assessed the sensibility of the derivations and theory.

**Review Assessment: Checking Correctness Of Experiments:**

I carefully checked the experiments.

**Review Assessment: Thoroughness In Paper Reading:**

I read the paper at least twice and used my best judgement in assessing the paper.

---

> ### Author Response · Authors · 2019-11-13
> **Answer**
>
> We thank the reviewer for the appreciation of the paper and insightful comments, in particular concerning the experimental results.
>
> - The resulting noise is indeed added in the image and in the gradient. The different noises come from the non deterministic nature of quantum procedures such as estimating the amplitude of a quantum state, or random outcomes of a measurement. Part of our work was to map these quantum errors to a resulting classically interpretable noise in the neural network itself (layers and gradients), in order to perform classical simulations. The fact that we apply a « Normal CNN » afterwards is the main goal of our algorithm, which is to reproduce the classical algorithm with quantum procedures to gain speedup.
>
> - The reviewer’s remarks concerning weight sharing and invariant representation is very interesting and will certainly be one of our focus for the future simulations. We believe that, in principle, properties of invariant representation, due to weight sharing in the convolution layer, are preserved since our quantum algorithm performs the same operations (convolution product, pooling). It is a good question to see if the noise could have a negative impact on this. We will look for a specific dataset and a relevant metric to quantify this property in comparison to classical CNN.
>
> - We are currently simulating the QCNN on larger and different datasets (e.g. CIFAR-10). A good advice from another reviewer will surely help us to perform these simulations more efficiently.

---

> > ### Comment · AnonReviewer3 · 2019-11-14
> > **Answer to authors**
> >
> > Concerning the addition of quantum noise, I'm a bit worried that there is no error bars on the learning curves since the noise is by definition stochastic. So, are these results really significant?
> >
> > I'm still not super convinced that the testing procedure is really demonstrating the announced performance gain of the method. I think it is a nice first step but it misses some of the important features of CNNs  (invariance, weight sharing etc.) and this is actually acknowledged in the answer. Without these features, my feeling is that the impact is quite limited.

---

> > > ### Author Response · Authors · 2019-11-15
> > > **Answer**
> > >
> > > We thank the reviewer for the insightful comments.
> > >
> > > We would like to clarify that our work provides a rigorous theoretical analysis showing that our quantum CNN is a faster and noise-robust adaptation of the classical CNN. As the quantum algorithm performs the same operations as the classical CNN (convolution, pooling, non-linearity), one expects that properties of classical CNN like invariance and weight sharing to be preserved. This is also evidenced so far by the experiments that show that even with the added noise the CNN converges fast and to a high accuracy model. We are performing further experiments in bigger data sets (which has been made possible through a comment of another referee) in order to further validate these properties. Nevertheless, both the theoretical analysis and the preliminary experiments strongly suggest that these properties will continue to hold.  We want to be very careful not to over-interpret these results as they don’t fully prove or disprove the capabilities of our quantum algorithm (which can only be done when the quantum hardware arrives), however they provide the best possible way of benchmarking the quantum algorithms in the present time.
> > >
> > > Concerning the error bars in the simulations, we have repeated these experiments many times and observed similar convergence. We will add the error bars in the final version.

---

### Official Review · AnonReviewer2 · 2019-10-23
**Official Blind Review #2**

**Rating:** 8

**Review:**

The authors present a quantum algorithm for approximating the forward pass and gradient computation of a classical convolutional neural network layer with pooling and a bounded rectifier activation. This algorithm has complexity bounds that would open up (for instance) the possibility of exponentially large filter banks, and the authors show through a simple, classical simulation approach that the resulting network is also likely to be trainable.

Feedback:

A few typos/formatting issues:
- The title accidentally includes "Conference Submissions"
- The in-text citation format frequently has the parentheses in the wrong place; this is surprisingly distracting!

Preliminaries:
- Maybe explain what the ith vector in the standard basis is in terms of |0> and |1>? I assume the answer is along the lines of |000>, |001>, |010>, etc.?

Main results:
- The sentence "a speedup compared to the classical CNN for both the forward pass and for training using backpropagation in certain cases" is ambiguous; does "in certain cases" qualify only training speed or also forward pass speed?

- There's a clear separation of background (which is concise and well explained) and contributions, but maybe it would be worth connecting the introduced algorithm more closely to existing work in non-convolutional quantum neural networks?

- Can you briefly justify (or cite) the claim that "most of the non linear functions in the machine learning literature can be implemented using small sized boolean circuits"?

- I'm a little confused about the discussion of quantum importance sampling on page 4. Could you give some intuition for the relationship between eta and the fraction of output values that are on average flushed to zero (is this 1 minus sigma?), and perhaps connect this to the literature about activation pruning and sparse NNs?

- Maybe define what you mean by "tomography" for ML folks without the quantum background?

- I'm convinced by the simulations, even though I shouldn't really be convinced by anything on MNIST... It just seems like the perturbations you're applying are all things that modern neural networks take in stride.

- The discussion of using a sigma-based classical sampling rather than the eta-based quantum importance sampling mentions a "Section C.1.15" which does not exist (I think you mean the end of Section C.1.5).

- Re: "We will use this analogy in the numerical simulations (Section 6) to estimate, for a particular QCNN architecture and a particular dataset of images, which values of σ are enough to allow the neural network to learn." My understanding is that you're getting empirical estimates of which values of sigma are enough; it would be valuable to convert those to estimates of which values of eta would be enough (given quantum networks of the size used in the classical simulation experiment, or given larger networks).

- The sampling procedure based on sigma might be inefficient in your PyTorch implementation, but it's certainly something that GPUs are fairly well suited to computing. There might be other PyTorch operators that would help here (perhaps Bernoulli sampling?) or if nothing else you could write a small custom CUDA kernel.

**Experience Assessment:**

I do not know much about this area.

**Review Assessment: Checking Correctness Of Derivations And Theory:**

I did not assess the derivations or theory.

**Review Assessment: Checking Correctness Of Experiments:**

I assessed the sensibility of the experiments.

**Review Assessment: Thoroughness In Paper Reading:**

I read the paper at least twice and used my best judgement in assessing the paper.

---

> ### Author Response · Authors · 2019-11-13
> **Answer**
>
> We thank the reviewer for the appreciation of the paper and insightful comments.
>
> - The reviewer is right concerning the meaning of the quantum state $|i>$ being the $i^{th}$ vector in the standard basis. If accepted, we will make the effort of introducing basic concepts of quantum computing to allow a clearer understanding of our work to the audience. As well, we will introduce more intuitively the concept of quantum tomography, namely the family of procedures that allow to retrieve a classical description of a quantum state by repeated measurements (and infer the values of the quantum amplitudes from the resulting distribution).
>
> - Our work in quantum deep learning is indeed related to previous works in quantum neural network cited in our paper, in particular the fully connected quantum neural network of Allcock et al. (2018). Their layer method is similar to ours and indeed could be explained in the appendix.
>
> - The speedup « in certain cases » concerns indeed both the forward pass and the whole training. We will change this sentence and be more explicit.
>
> - Applying a non linearity (as ReLu activation function) in a quantum circuit is a difficult challenge. In our solution, once the value is encoded as a bit string in a quantum register, we can apply a non linearity on it. The circuit that modifies the value accordingly depends on the non linearity considered. For ReLu or other positive simple rules (piecewise linear functions, indicator functions), one could imagine a simple circuit involving few gates to act on the bit strings. Most importantly, the size of such circuits will have a constant depth that doesn’t depend on the algorithm parameters. In our sentence, « boolean » refers to a classical and explicit series of gates. Note however that implementing more complex non linearities such as tanh could imply a taylor decomposition of the function in order to approximate it with a small number of gates. This explains our choice of the ReLu function.
>
> - Our « quantum importance sampling » can be parametrized in two manners: $\eta$ that relates to the precision of the tomography, or $\sigma$ that corresponds to the ratio of elements sampled (the others being set to zero). The relation between the two approaches is given in Appendix, Section C.1.5, namely we have $\sigma = N/\eta^2$ where $N$ is the size of the output image. We agree that the explanations could be clearer and we will make the effort to present it better. In our opinion, the Sigma perspective is more intuitive when considering image processing (as shown in Figure 1), and more explainable than Eta which implicitly depends on the size $N$.
>
> - This particular sampling described above is purely a quantum effect and has no known classical usage or reason to be. In a way, it can be seen to a non deterministic activation function. It is reducing the number of non zero values in the layers themselves, and not (directly) in the weights of the kernels. Therefore it might, or not, be related to pruning, drop out or sparse NN. We appreciate this comment and will research on that analogy.
>
> - We thank the reviewer for the advice concerning the PyTorch implementation. We will certainly use this to perform further simulations on different and larger datasets. This could help us save a lot of time.
>
> - All remarks concerning typos and formatting will be taken into account in the final version.

---

### Official Review · AnonReviewer1 · 2019-10-29
**Official Blind Review #1**

**Rating:** 8

**Review:**

The authors provide a comprehensive study, with theory and classical simulation of the quantum system, on how to increaese the speed of CNN inference and training using qubits. They proved an intrigued compilation of a quantized convolutional system.

For an audiance who are not quantum experts one could clarify a few properties of the described "quantization".  It is not able to take multiple images in at the same time as quantum superpositions. Sometimes the expectence of a quantum machine leanring is that it would train the system at a single instance. Here the increase in effciency comes from being able to perform marix multiplication in quantum realm.

 The manuscripd describes a creative way to bring bolean operartions-defined non-linearities into quantum neural networks, at the cost of having to force the system back to classical domain at each layer - and then encoding it back to qubits for the next layer operations. The price has to payed as unitary operators  (the ones that preserve entaglement) are inherently linear.

Remarks:
The authors should point out how this is specific to convolutional neural networks. It looks to me that the same algorithm could be used for fully connected or even attention based systems, as it is just a matrix multiplication as well. Anyway, the manuscript provides the take into account the steps required specifically for a CNN.

Some minor remarks:
For classical systems the capped Relu is inferior as it reduces the range of values of activations where there is driving force. Sometimes one is using a parametrized version of ReLU that has a small positive slope for negative values.
It is not clear to me how this would not be the case with a quantum implementions. In your simulations, does the value of the saturation constant C, change the speed of convergence to a good solutions.

The capped Relu reminds me a lot of a Tanh non-linearity  that works well for LSTMs but are not very good for CNNs.

I would suggest that for the conference presentation the authors try to bring out the essential within a less formal setting to open it to a wider ML audience.  For the manuscript the level is good with a proper use of appendix to shorten the main narrative.



**Experience Assessment:**

I have published in this field for several years.

**Review Assessment: Checking Correctness Of Derivations And Theory:**

I assessed the sensibility of the derivations and theory.

**Review Assessment: Checking Correctness Of Experiments:**

I assessed the sensibility of the experiments.

**Review Assessment: Thoroughness In Paper Reading:**

I read the paper at least twice and used my best judgement in assessing the paper.

---

> ### Author Response · Authors · 2019-11-13
> **Answer**
>
> We thank the reviewer for the appreciation of the paper and insightful comments.
>
> - If accepted, we will do our best effort to ensure an appropriate and clear presentation of quantum machine learning. In particular we will communicate to the audience the core principles of quantum computing and main methods to « quantize » machine learning algorithms (quantum vectors, quantum linear algebra, quantum distance estimation). As well, we will indicate clearly the benefit and limitations of such methods.
>
> - Concerning the remark on the potentiel limitations of a capped ReLu activation function, we agree with the reviewer and will pursue further simulations to quantify the implications on the accuracy and training time.

---

### Official Review · AnonReviewer4 · 2019-10-31
**Official Blind Review #4**

**Rating:** 6

**Review:**

This submission proposed a quantum convolutional neural network (QCNN). The theoretical results in section 3 state the existence of the QCNN satisfying certain conditions. The QCNN is given by sections 4 and 5, with empirical evaluates in section 6. This subject is out of my usual area. However, I tend to think this subject is interesting to the ICLR audience due to the recent advancement in quantum computing.

title, remove "conference submissions"

Section 2, introduce QRAM.

Somewhere around section 2-3, and in section 6, it has to be mentioned whether the proposed QCNN requires special hardware, and what is the hardware, and why it is required.

Note the cited Cong et al. (2018) has been published in nature physics. As you both used the term "QCNN", it is better to explain more clearly what is the main difference in the main text.

**Experience Assessment:**

I do not know much about this area.

**Review Assessment: Checking Correctness Of Derivations And Theory:**

I assessed the sensibility of the derivations and theory.

**Review Assessment: Checking Correctness Of Experiments:**

I assessed the sensibility of the experiments.

**Review Assessment: Thoroughness In Paper Reading:**

I made a quick assessment of this paper.

---

> ### Author Response · Authors · 2019-11-13
> **Answer**
>
> We thank the reviewer for the appreciation of the paper and good remarks.
>
> - If accepted, we will make our best effort to introduce the concept of QRAM, briefly in the main paper and in details in the appendix. The reviewer is right to request this as it is a very important aspect of quantum machine learning and quantum computing in general.
>
> - The previous paper of Cong et al. «Quantum Convolution Neural Network» published in Nature Physics is an excellent contribution, but is conceptually different from our submission for the following reasons: Cong et al. have defined a new quantum circuit model (inspired from quantum physics), with properties that can be applied for signal processing (eventually image if generalized to 2D inputs). Their algorithm can be used for learning or classifying phase of quantum physical systems, and shares similar aspects with a CNN, hence the naming. However, they are not reproducing the precise operations that a classical CNN is performing (convolution product, non linearities, pooling, backpropagation by gradient descent, etc.) layer by layer, which is the topic of our work. In other words, we are defining the first quantum circuit simulating a classical CNN. The reviewer is right to notice this and we will explain the specificity of each paper to avoid any ambiguity.
>
> - The quantum algorithm described in our work is « hardware agnostic » in the sense that it is built on a theoretical and universal set of quantum gates (e.g. Control-NOT, Hadamard, Rotations) that should be implementable by any hardware (superconducting circuits, photonics, ions etc.). If accepted, we will precise and explain this core fact in the final version.
>
> - We will promptly remove the undesired « conference submission » in the title.

---

### Decision · Program_Chairs · 2019-12-19

**Decision:**

Accept (Poster)

**Comment:**

Four reviewers have assessed this paper and they have scored it as 6/6/6/6 after rebuttal. Nonetheless, the reviewers have raised a number of criticisms and the authors are encouraged to resolve them for the camera-ready submission. Especially, the authors should take care to make this paper accessible (understandable) to the ML community as ICLR is a ML venue (rather than quantum physics one). Failure to do so will likely discourage the generosity of reviewers toward this type of submissions in the future.